# THE IN-SAMPLE SOFTMAX FOR OFFLINE REINFORCEMENT LEARNING

**Chenjun Xiao**[*,1,2], **Han Wang**[*,†,2], **Yangchen Pan**[†,3], **Adam White**[2], **Martha White**[2]

[1] Huawei Noah's Ark Lab
[2] University of Alberta; Alberta Machine Intelligence Institute (Amii)
[3] University of Oxford

## ABSTRACT

Reinforcement learning (RL) agents can leverage batches of previously collected data to extract a reasonable control policy. An emerging issue in this offline RL setting, however, is that the bootstrapping update underlying many of our methods suffers from insufficient action-coverage: standard max operator may select a maximal action that has not been seen in the dataset. Bootstrapping from these inaccurate values can lead to overestimation and even divergence. There are a growing number of methods that attempt to approximate an *in-sample* max, that only uses actions well-covered by the dataset. We highlight a simple fact: it is more straightforward to approximate an in-sample *softmax* using only actions in the dataset. We show that policy iteration based on the in-sample softmax converges, and that for decreasing temperatures it approaches the in-sample max. We derive an In-Sample Actor-Critic (AC), using this in-sample softmax, and show that it is consistently better or comparable to existing offline RL methods, and is also well-suited to fine-tuning. We release the code at github.com/hwang-ua/inac_pytorch.

## 1 INTRODUCTION

A common goal in reinforcement learning (RL) is to learn a control policy from data. In the offline setting, the agent has access to a batch of previously collected data. This data could have been gathered under a near-optimal behavior policy, from a mediocre policy, or a mixture of different policies (perhaps produced by several human operators). A key challenge is to be robust to this data gathering distribution, since we often do not have control over data collection in some application settings. Most approaches in offline RL learn action-values, either through Q-learning updates—bootstrapping off of a maximal action in the next state—or for actor-critic algorithms where the action-values are updated using temporal-difference (TD) learning updates to evaluate the actor.

In either case, poor action coverage can interact poorly with bootstrapping, yielding bad performance. The action-value updates based on TD involves bootstrapping off an estimate of values in the next state. This bootstrapping is problematic if the value is an overestimate, which is likely to occur when there are actions that are never sampled in a state (Fujimoto et al., 2018; Kumar et al., 2019; Fujimoto et al., 2019). When using a maximum over actions, this overestimate will be selected, pushing up the value of the current state and action. Such updates can lead to poor policies and instability (Fujimoto et al., 2018; Kumar et al., 2019; Fujimoto et al., 2019).

There are two main approaches in offline RL to handle this over-estimation issue. One direction constrains the learned policy to be similar to the dataset policy (Wu et al., 2019; Peng et al., 2020; Nair et al., 2021; Brandfonbrener et al., 2021; Fujimoto & Gu, 2021). A related idea is to constrain the stationary distribution of the learned policy to be similar to the data distribution (Yang et al., 2022). The challenge with both these approaches is that they rely on the dataset being generated by an expert or near-optimal policy. When used on datasets from more suboptimal policies—like those commonly found in industry—they do not perform well (Kostrikov et al., 2022). The other approach is bootstrap off pessimistic value estimates (Kidambi et al., 2020; Kumar et al., 2020; Kostrikov et al.,

---

[*]These authors contributed equally to this work
[†]Work was done while the author was at Huawei Noah's Ark Lab
`{chenjun, han8, amw8, whitem}@ualberta.ca; yangchen.pan@eng.ox.ac.uk`

2021; Yu et al., 2021; Jin et al., 2021; Xiao et al., 2021) and relatedly to identify and reduce the influence of out-of-distribution actions using ensembles (Kumar et al., 2019; Agarwal et al., 2020; Ghasemipour et al., 2021; Wu et al., 2021; Yang et al., 2021; Bai et al., 2022).

One simply strategy that has been more recently proposed is to constrain the set of actions considered for bootstrapping to the support of the dataset $\mathcal{D}$. In other words, if $\pi_{\mathcal{D}}(a|s)$ is the conditional action distribution underlying the dataset, then we use $\max_{a':\pi_{\mathcal{D}}(a'|s')>0} q(s', a')$ instead of $\max_{a'} q(s', a')$: a constrained or *in-sample max*. This idea was first introduced for Batch-Constrained Q-learning (BCQ) (Fujimoto et al., 2019) in the tabular setting, with a generative model used to approximate and sample $\pi_{\mathcal{D}}(a|s)$ (Fujimoto et al., 2019; Zhou et al., 2020; Wu et al., 2022). Implicit Q-learning (IQL) (Kostrikov et al., 2022) was the first model-free approximation to use this in-sample max, with a later modification to be less conservative (Ma et al., 2022). IQL instead uses expectile regression, to push the action-values to predict upper expectiles that are a (close) lower bound to the true maximum. The approach nicely avoids estimating $\pi_{\mathcal{D}}$, and empirically performs well. Using only actions in the dataset is beneficial, because it can approach is be difficult to properly constrain the support of the learned model for $\pi_{\mathcal{D}}$ and ensure it does not output out-of-distributions actions.

There are, however, a few limitations to IQL. The IQL solution depends on the action distribution not just the support. In practice, we would expect IQL to perform poorly when the data distribution is skewed towards suboptimal actions in some states, pulling down the expectile regression targets. We find evidence for this in our experiments. Additionally, convergence is difficult to analyze because expectile regression does not have a closed-form solution. One recent work showed that the Bellman operator underlying an expectile value learning algorithm is a contraction, but only for the setting with deterministic transitions (Ma et al., 2022).

In this work, we revisit how to directly use the in-sample max. Our key insight is simple: sampling under support constraints is more straightforward for the softmax, in the entropy-regularized setting. We first define the *in-sample softmax* and show that it maintains the same contraction and convergence properties as the standard softmax. Further, we show that with a decreasing temperature (entropy) parameter, the in-sample softmax approaches the in-sample max. This formulation, therefore, is both useful for those wishing to incorporate entropy-regularization and to give a reasonable approximation to the in-sample max by selecting a small temperature. We then show that we can obtain a policy update that relies primarily on sampling from the dataset—which is naturally in-sample—rather than requiring samples from an estimate of $\pi_{\mathcal{D}}$. We conclude by showing that our resulting In-sample Actor-critic algorithm consistently outperforms or matches existing methods, despite being a notably simpler method, in offline RL experiments with and without fine-tuning.

## 2 PROBLEM SETTING

In this section we outline the key issue of action-coverage in offline RL that we address in this work.

### 2.1 MARKOV DECISION PROCESS

We consider finite Markov Decision Process (MDP) determined by $M = \{\mathcal{S}, \mathcal{A}, P, r, \gamma\}$ (Puterman, 2014) , where $\mathcal{S}$ is a finite state space, $\mathcal{A}$ is a finite action space, $\gamma \in [0, 1)$ is the discount factor, $r : \mathcal{S} \times \mathcal{A} \to \mathbb{R}$ and $P : \mathcal{S} \times \mathcal{A} \to \Delta(\mathcal{S})$ are the reward and transition functions.[1] The *value function* specifies the future discounted total reward obtained by following a policy $\pi : \mathcal{S} \to \Delta(\mathcal{A})$, $v^\pi(s) = \mathbb{E}^\pi[\sum_{t=0}^\infty \gamma^t r(s_t, a_t)|s_0 = s]$ where we use $\mathbb{E}^\pi$ to denote the expectation under the distribution induced by the interconnection of $\pi$ and the environment. The corresponding *action-value* function is $q^\pi(s, a) = r(s, a) + \gamma \mathbb{E}_{s' \sim P(\cdot|s,a)}[v^\pi(s')]$. There exists an *optimal policy* $\pi^*$ that maximizes the values for all states $s \in \mathcal{S}$. We use $v^*$ and $q^*$ to denote the optimal value functions. The optimal value satisfies the *Bellman optimality equation*,

$$v^*(s) = \max_a r(s, a) + \gamma \mathbb{E}_{s'}[v^*(s')], \quad q^*(s, a) = r(s, a) + \gamma \mathbb{E}_{s' \sim P(\cdot|s,a)}\left[\max_{a'} q^*(s', a')\right]. \quad (1)$$

In this work we more specifically consider the *entropy-regularized MDP* setting—also called the maximum entropy setting—where an entropy term is added to the reward to encourage the policy to

---

[1]We use the standard notation $\Delta(\mathcal{X})$ to denote the set of probability distributions over a finite set $\mathcal{X}$.

be stochastic. The maximum-entropy value function is defined as

$$\tilde{v}^\pi(s) = v^\pi(s) + \tau\mathbb{H}(s,\pi)\,, \quad \mathbb{H}(s,\pi) = \mathbb{E}^\pi\left[\sum_{t=0}^{\infty} -\gamma^t \log\pi(a|s)\Big|s_0 = s\right]\,, \tag{2}$$

for temperature $\tau$ and $\mathbb{H}$ the *discounted entropy regularization*. The corresponding maximum-entropy action-value function is $\tilde{q}^\pi(s,a) = r(s,a) + \gamma\mathbb{E}_{s'\sim P(s,a)}[\tilde{v}^\pi(s')]$, with soft Bellman optimality equations similarly modified as described in the next section. As $\tau \to 0$, we recover the original value function definitions. The entropy-regularized setting has become widely used (Ziebart et al., 2008; Mnih et al., 2016; Nachum et al., 2017; Asadi & Littman, 2017; Haarnoja et al., 2018; Mei et al., 2019; Xiao et al., 2019), because it 1) encourages exploration (Ziebart et al., 2008), 2) often makes objectives more smooth (Mei et al., 2019), and 3) provides these improvements even with small temperatures that do not significantly bias the solution to the original MDP (Song et al., 2019).

## 2.2 OFFLINE REINFORCEMENT LEARNING

In this work, we consider the problem of learning an optimal decision making policy from a previously collected offline dataset $\mathcal{D} = \left\{s_i, a_i, r_i, s_i'\right\}_{i=0}^{n-1}$. We assume that the data is generated by executing a *behavior policy* $\pi_\mathcal{D}$. Note that we do assume direct access to $\pi_\mathcal{D}$. In offline RL, the learning algorithm can only learn from samples in this $\mathcal{D}$ without further interaction with the environment.

One primary issue in offline RL is that $\pi_\mathcal{D}$ may not have full coverage over actions. Greedy decisions based on a learned value $q \approx q^*$ could be problematic, especially when the value is an overestimate for out-of-distribution actions (Fujimoto et al., 2019). To overcome this issue, one popular approach is to constrain the learned policy to be similar to $\pi_\mathcal{D}$, such as by adding a KL-divergence term: $\max_\pi \mathbb{E}_{s\sim\rho}[\sum_a \pi(a|s)q(s,a) - \tau D_{\mathrm{KL}}(\pi(\cdot|s)||\pi_\mathcal{D}(\cdot|s))]$ for some $\tau > 0$. The optimal policy for this objective must be on the support of $\pi_\mathcal{D}$: the KL constraint makes sure $\pi(a|s) = 0$ as long as $\pi_\mathcal{D}(a|s) = 0$. This optimal policy, with closed-form solution $\pi'(a|s) \propto \pi_\mathcal{D}(a|s)\exp(q(s,a)/\tau)$, is also guaranteed to be an improvement on $\pi_\mathcal{D}$. Many offline RL algorithms are based on this nice idea (Wu et al., 2019; Peng et al., 2020; Nair et al., 2021; Brandfonbrener et al., 2021; Fujimoto & Gu, 2021).[2] This KL constraint, however, can result in poor $\pi'$ when $\pi_\mathcal{D}$ is sub-optimal, confirmed both in previous studies (Kostrikov et al., 2022) and our experimental results.

The other strategy is to consider an in-sample policy optimization, $\max_{\pi \preceq \pi_\mathcal{D}} \sum_{a\in\mathcal{A}} \pi(a|s)q(s,a)$, where $\pi \preceq \pi_\mathcal{D}$ indicates the support of $\pi$ is a subset of $\pi_\mathcal{D}$. This approach more directly avoids selecting out-of-distribution actions. Though a simple idea, approximating this with a simple algorithm has been elusive, as discussed above. The simplest idea is to estimate $\pi_\omega \approx \pi_\mathcal{D}$ and directly constrain the support by sampling candidate actions from $\pi_\omega$, as proposed for Batch-Constrained Q-learning (Fujimoto et al., 2019). This simple approach, however, may not avoid bootstrapping from out-of-sample actions due to the error in the estimate $\pi_\omega$.

Surprisingly, the small modification to the *in-sample softmax* (Section 3) has not yet been considered for offline RL. Yet, moving from the in-sample (hard) max to the in-sample softmax facilitates developing a simple algorithm, as we discuss in the remainder of this work.

# 3 THE IN-SAMPLE SOFTMAX OPTIMALITY

This section introduces the in-sample softmax optimality that provides a simple implementation of in-sample bootstrapping. We first describe the standard soft Bellman optimality equations, then the modification to consider in-sample bootstrapping. Our simple algorithm comes from stepping back and recognizing the utility of considering in-sample bootstrapping for the entropy-regularized setting rather than only for the hard-max.

The soft Bellman optimality equations for maximum-entropy RL use the softmax in place of the max,

$$\tilde{q}^*(s,a) = r(s,a) + \gamma\mathbb{E}_{s'\sim P(\cdot|s,a)}\left[\tau\log\sum_{a\in\mathcal{A}} e^{\tilde{q}^*(s',a')/\tau}\right]. \tag{3}$$

---

[2] Fujimoto & Gu (2021) use a *behavior cloning regularization* $(\pi(s) - a)^2$, where $a$ is action in the dataset. We note it is exactly a KL regularization under Gaussian parameterization with standard deviation. Brandfonbrener et al. (2021) propose a *one-step* policy improvement method: first learn the value of $\pi_\mathcal{D}$, then directly train a policy to maximize the learned value. Thus this is indeed a behavior regularized approach.

This comes from the fact that hard max with entropy regularization is $\max_{p \in \Delta(\mathcal{A})} \sum_{a \in \mathcal{A}} p(a)q(s,a) + \tau \mathbb{H}(p) = \tau \log \sum_{a \in \mathcal{A}} e^{q(s,a)/\tau}$. As $\tau \to 0$, softmax (log-sum-exp) approaches the max.[3]

We can modify this update to restrict the softmax to the support of $\pi_{\mathcal{D}}$:

$$\tilde{q}^*_{\pi_{\mathcal{D}}}(s,a) = r(s,a) + \gamma \mathbb{E}_{s' \sim P(\cdot|s,a)} \left[ \tau \log \sum_{a': \pi_{\mathcal{D}}(a'|s')>0} e^{\tilde{q}^*_{\pi_{\mathcal{D}}}(s',a')/\tau} \right]. \tag{4}$$

We call Eq. (4) the *in-sample softmax optimality equation*. It is interesting to note that we can use a simple reformulation that facilitates sampling the inner term. For any $q$,

$$\begin{aligned}
\sum_{a: \pi_{\mathcal{D}}(a|s)>0} e^{q(s,a)/\tau} &= \sum_{a: \pi_{\mathcal{D}}(a|s)>0} \pi_{\mathcal{D}}(a|s)\pi_{\mathcal{D}}(a|s)^{-1} e^{q(s,a)/\tau} \\
&= \sum_{a: \pi_{\mathcal{D}}(a|s)>0} \pi_{\mathcal{D}}(a|s) e^{-\log \pi_{\mathcal{D}}(a|s)} e^{q(s,a)/\tau} \\
&= \mathbb{E}_{a \sim \pi_{\mathcal{D}}(\cdot|s)} \left[ e^{q(s,a)/\tau - \log \pi_{\mathcal{D}}(a|s)} \right]. \tag{5}
\end{aligned}$$

This reformulation does not perfectly remove the role of $\pi_{\mathcal{D}}(a|s)$, but it is significantly reduced. The support is no longer constrained using $\pi_{\mathcal{D}}$ and instead the values are simply shifted by this term involving $\pi_{\mathcal{D}}$. We will use this strategy below to develop our algorithm.

There are a few interesting facts to note about in-sample softmax. First, we can show that similarly to the standard maximum-entropy bootstrap (shown formally in Lemma 3), we have for any $q$

$$\tau \log \sum_{a: \pi_{\mathcal{D}}(a|s)>0} e^{q(s,a)/\tau} = \max_{\pi \preceq \pi_{\mathcal{D}}} \sum_a \pi(a|s)q(s,a) + \tau \mathbb{H}(\pi). \tag{6}$$

Though this outcome is intuitive, it is a nice property that restricting the support of the log-sum-exp maintains the same relationship to the maximum-entropy update with the same support constraint. It extends this result for the soft Bellman optimality update to the setting with a support constraint. From this perspective, in-sample softmax can also be viewed as a tool for *conservative exploration*: exploring to prevent getting stuck in a local optima, while still being suspicious of what the data does not know. This is especially important when $q$ is a learned value approximation.

Second, we can also obtain a closed-form greedy policy using the above (shown formally in Lemma 3), which we call the *in-sample softmax greedy policy*: for any $q$,

$$\pi_{\pi_{\mathcal{D}},q}(a|s) \propto \pi_{\mathcal{D}}(a|s) \exp\left( \frac{q(s,a)}{\tau} - \log \pi_{\mathcal{D}}(a|s) \right), \tag{7}$$

This closed-form solution looks similar to the KL-regularized solution mentioned in Section 2.2, where $\pi'$ is constrained to be similar to $\pi_{\mathcal{D}}$. The only difference is the additional $-\log \pi_{\mathcal{D}}$ term inside the exponential. This small difference, however, has a big impact. It allows the resulting policy to deviate much more from $\pi_{\mathcal{D}}$. In fact, because $\exp(-\log \pi_{\mathcal{D}}(a|s)) = \pi_{\mathcal{D}}(a|s)^{-1}$, the above is equivalent to $\pi_{\pi_{\mathcal{D}},q}(a|s) = 0$ when $\pi_{\mathcal{D}}(a|s) = 0$ and otherwise $\pi_{\pi_{\mathcal{D}},q}(a|s) \propto \exp(q(s,a)/\tau)$[4]. The new policy $\tilde{\pi}_{\pi_{\mathcal{D}}}$ is not skewed by the action probabilities in $\pi_{\pi_{\mathcal{D}},q}$; it just has the same support.

## 4    THEORETICAL CHARACTERIZATION OF IN-SAMPLE SOFTMAX

In this section we prove in-sample softmax maintains the convergence properties of the standard softmax. In particular, Bellman updates with the in-sample softmax are convergent, and the resulting in-sample softmax optimal policy approaches the in-sample optimal policy as we reduce the temperature to zero. All proofs are given in Appendix A.

---

[3]Note that this softmax operator for the soft Bellman optimality equation is different from the softmax Bellman operator, which uses an expectation in the bootstrap over a softmax policy and which is know to have issue with not being an contraction (Asadi & Littman, 2017). The log-sum-exp formula is a standard way to approximate the max, and is naturally called a softmax.

[4]We define $0 \cdot \infty = 0$

We can contrast our in-sample softmax optimality equation in (4) to the *in-sample Bellman optimality equation* introduced by Fujimoto et al. (2019) for the hard max,

$$q^*_{\pi_\mathcal{D}}(s,a) = r(s,a) + \gamma\mathbb{E}_{s'\sim P(\cdot|s,a)}\left[\max_{a':\pi_\mathcal{D}(a'|s')>0} q^*_{\pi_\mathcal{D}}(s',a')\right]. \tag{8}$$

We first show that the policy produced using the in-sample softmax optimality equation is a good approximation to that given by the in-sample Bellman optimality equation.

**Theorem 1.** *Let $\tilde{q}^*_{\pi_\mathcal{D}}$ be the in-sample softmax optimal value function. We have $\lim_{\tau\to 0}\tilde{q}^*_{\pi_\mathcal{D}} = q^*_{\pi_\mathcal{D}}$. Moreover, let $\mathbb{I}$ be an indicator function and $\pi_{\pi_\mathcal{D}}(a|s) = \mathbb{I}(a = \arg\max_{a:\pi_\mathcal{D}(a)>0} q^*_{\pi_\mathcal{D}}(s,a))$ be the in-sample optimal policy w.r.t $q^*_{\pi_\mathcal{D}}$. Define the in-sample softmax optimal policy,*

$$\tilde{\pi}^*_{\pi_\mathcal{D}}(a|s) \propto \pi_\mathcal{D}(a|s)\exp\left(\frac{\tilde{q}^*_{\pi_\mathcal{D}}(s,a)}{\tau} - \log\pi_\mathcal{D}(a|s)\right). \tag{9}$$

*We have $\lim_{\tau\to\infty}\tilde{\pi}^*_{\pi_\mathcal{D}} = \pi^*_{\pi_\mathcal{D}}$.*

Now we show that we can reach the in-sample softmax optimal solution, using either value iteration or policy iteration. For value iteration, we define the *in-sample softmax optimality operator*

$$(\mathcal{T}_{\pi_\mathcal{D}}q)(s,a) = r(s,a) + \gamma\mathbb{E}_{s'\sim P(\cdot|s,a)}\left[\tau\log\sum_{a':\pi_\mathcal{D}(a'|s')>0} e^{q(s',a')/\tau}\right]. \tag{10}$$

The next result shows that $\mathcal{T}_{\pi_\mathcal{D}}$ is a contraction, and therefore *in-sample soft value iteration*, using $q_{t+1} = \mathcal{T}_{\pi_\mathcal{D}}q_t$, is guaranteed to converge to the in-sample softmax optimal value in the tabular case.

**Theorem 2.** *For $\gamma < 1$, the fixed point of the in-sample softmax optimality operator exists and is unique. Thus, in-sample soft value iteration converges to the in-sample softmax optimal value $\tilde{q}^*_{\pi_\mathcal{D}}$.*

As highlighted by Equation (6), the in-sample softmax policy corresponds to the solution of the maximum entropy policy optimization. This implies that similarly to Soft Actor-Critic (Haarnoja et al., 2018), we can apply policy iteration to find this policy. Let $\pi_t$ be the policy at iteration $t$. The algorithm first learns the value function $\tilde{q}^{\pi_t}$, then updates the policy $\pi_{t+1}$ such that $\tilde{q}^{\pi_t} \leq \tilde{q}^{\pi_{t+1}}$. The following result shows that this procedure guarantees policy improvement.

**Lemma 1.** *Let $\pi_t$ be a policy such that $\pi_t \preceq \pi_\mathcal{D}$. Define*

$$\pi_{t+1}(a|s) \propto \pi_\mathcal{D}(a|s)\exp\left(\frac{\tilde{q}^{\pi_t}(s,a)}{\tau} - \log\pi_\mathcal{D}(a|s)\right). \tag{11}$$

*Then $\pi_{t+1} \preceq \pi_\mathcal{D}$ and $\tilde{q}^{\pi_{t+1}} \geq \tilde{q}^{\pi_t}$.*

Note that $\pi_{t+1}$ not only ensures policy improvement, but also stays in the support of $\pi_\mathcal{D}$. Now let us define the *on-policy entropy-regularized operator*,

$$(\mathcal{T}^\pi q)(s,a) = r(s,a) + \gamma\mathbb{E}_{s',a'\sim P^\pi(\cdot|s,a)}[q(s',a') - \tau\log\pi(a'|s')]. \tag{12}$$

Since this operator is a contraction (shown formally in Lemma 5), we can evaluate $\tilde{q}^\pi$ by repeatedly applying $\mathcal{T}^\pi q$ from any $q$ until converge. These updates give rise to the *in-sample soft policy iteration* algorithm that iteratively updates the policy using (11) and evaluate its by using $\mathcal{T}^\pi$. The convergence for the tabular case is given below.

**Theorem 3.** *For $\gamma < 1$, starting from any initial policy $\pi$ such that $\pi \preceq \pi_\mathcal{D}$, in-sample soft policy iteration converges to the in-sample softmax optimal policy $\tilde{\pi}^*_{\pi_\mathcal{D}}$.*

## 5 POLICY OPTIMIZATION USING THE IN-SAMPLE SOFTMAX

In this section, we develop an In-sample Actor-critic (AC) algorithm based on the in-sample softmax. This is the first time we see the utility of the in-sample softmax, to facilitate sampling actions from $\pi_\mathcal{D}$ using only actions in the dataset. This contrasts other direct methods, like BCQ that approximate the in-sample max by sampling from an approximate $\pi_\omega$ (Fujimoto et al., 2019). Throughout this section we generically develop the algorithm for continuous and discrete actions. Instead of using

sums, therefore, we primarily write formulas using expectations, which allow for either discrete or continuous actions.

The In-sample AC algorithm is similar to SAC (Haarnoja et al., 2018), except that we carefully consider out-of-sample actions. We similarly learn an actor $\pi_\psi$ with parameters $\psi$, action-values $q_\theta$ with parameters $\theta$ and a value function $v_\phi$ with parameters $\phi$. Additionally, we learn $\pi_\omega \approx \pi_\mathcal{D}$. We need this to define the greedy policy shown above in Equation (11), but do not directly use it to constrain the support over actions.

The first step in the algorithm is to extract $\pi_\omega \approx \pi_\mathcal{D}$. We do so using a simple maximum likelihood loss on the dataset: $\mathcal{L}_{\text{behavior}}(\omega) = -\mathbb{E}_{(s,a)\sim\mathcal{D}}[\log \pi_\omega(a|s)]$. We do not add any additional tricks to try to ensure action probabilities are zero where $\pi_\mathcal{D}(a|s) = 0$, because this $\pi_\omega$ only plays a smaller role in our update. It will only be used to adjust the greedy policy, and will only be queried on actions in the dataset.

Then we use a similar approach to SAC, where we alternate between estimating $q_\theta$ and $v_\phi$ for the current policy and improving the policy by minimizing a KL-divergence to the soft greedy policy. The main difference here to SAC is that we update towards the in-sample soft greedy. We cannot directly use Equation (11), which involves $\pi_\mathcal{D}$ in the update, but can replace $\pi_\mathcal{D}$ in the update with our approximation $\pi_\omega$. We therefore update towards an approximate in-sample soft greedy policy

$$\hat{\pi}_{\pi_\mathcal{D},q_\theta}(a|s) = \pi_\mathcal{D}(a|s)\exp\left(\frac{q_\theta(s,a) - Z(s)}{\tau} - \log\pi_\omega(a|s)\right)$$

where $Z(s) = \tau\log\int_a \pi_\mathcal{D}(a|s)\exp(\frac{q_\theta(s,a)}{\tau} - \log\pi_\omega(a|s))da$ is the normalizer to give a valid distribution. We minimize a forward KL to this in-sample soft greedy policy, because that allows us to sample the KL by only sampling actions from the dataset. To see why, notice that

$$D_{\text{KL}}(\hat{\pi}_{\pi_\mathcal{D},q_\theta}(\cdot|s)||\pi_\psi(\cdot|s)) = -\mathbb{E}_{a\sim\hat{\pi}_{\pi_\mathcal{D},q_\theta}(\cdot|s)}[\log\pi_\psi(a|s) - \log\hat{\pi}_{\pi_\mathcal{D},q_\theta}(a|s)] \qquad (13)$$

$$= \mathbb{E}_{a\sim\pi_\mathcal{D}(\cdot|s)}\left[\exp\left(\frac{q_\theta(s,a) - Z(s)}{\tau} - \log\pi_\omega(a|s)\right)(\log\pi_\psi(a|s) + \log\hat{\pi}_{\pi_\mathcal{D},q_\theta}(a|s))\right]$$

The expectation is now over samples $a \sim \pi_\mathcal{D}(\cdot|s)$; the actions in the dataset are precisely sampled from $\pi_\mathcal{D}$. To sample the gradient for this loss, we also need an estimate for $Z(s)$. We use our parameterized $v_\phi$ to estimate $Z$; we discuss why this is reasonable below. The final loss function for the actor $\pi_\psi$ is

$$\mathcal{L}_{\text{actor}}(\psi) = -\mathbb{E}_{s,a\sim\mathcal{D}}\left[\exp\left(\frac{q_\theta(s,a) - v_\phi(s)}{\tau} - \log\pi_\omega(a|s)\right)\log\pi_\psi(a|s)\right]. \qquad (14)$$

For the value function we use standard value function updates for the entropy-regularized setting. The objectives are

$$\mathcal{L}_{\text{baseline}}(\phi) = \mathbb{E}_{s\sim\mathcal{D},a\sim\pi_\psi(s)}\left[\frac{1}{2}\left(v_\phi(s) - (q_\theta(s,a) - \tau\log\pi_\psi(a|s))\right)^2\right] \qquad (15)$$

$$\mathcal{L}_{\text{critic}}(\theta) = \mathbb{E}_{s,a,r,s'\sim\mathcal{D}}\left[\frac{1}{2}\left(r + \gamma v_\phi(s') - q_\theta(s,a)\right)^2\right]. \qquad (16)$$

The action-values use the estimate of $v_\phi$ in the next state, and so avoids using out-of-distribution actions. The update to the value function, $v_\phi$, uses only actions sampled from $\pi_\psi$, which is being optimized to stay in-sample. Periodically, however, $v_\phi$ may bootstrap off of out-of-distribution actions because we do not guarantee that $\pi_\psi \preceq \pi_\mathcal{D}$. In fact, in early learning we expect $\pi_\psi$ will not satisfy this property. Despite this, the actor update will progressively reduce the probability of these out-of-distribution actions, even if temporarily the action-values overestimate their value, because the actor update pushes $\pi_\psi$ towards the in-sample greedy policy. This means that the overestimate is unlikely to significantly skew the actor, and progressively the overestimate should be reduced as the support of $\pi_\psi$ is reduced.

Finally, instead of learning a separate approximation for $Z$, we opt for the simpler approach of using $v_\phi$. The reason is that $v_\phi$ should provide a reasonable approximation to $Z$ because of the relationship between soft values and $Z$. From Equations (7) and (8) (formally proved in Lemma 3), we know that

the soft values for the in-sample soft greedy policy $\tilde{\pi}_{\pi_\mathcal{D}, q_\theta}$ correspond to the normalizer $Z$ for that policy. Therefore, given that $\pi_\omega \approx \pi_\mathcal{D}$, the soft values of approximate in-sample soft greedy policy $\hat{\pi}_{\pi_\mathcal{D}, q_\theta}$ should also be similar to $Z$. Since we optimize our policy to approximate $\hat{\pi}_{\pi_\mathcal{D}, q_\theta}$, we expect its entropy-regularized value, which is the learning target of $v_\phi$ as shown in Equation (15), to be a good approximation of $Z$.

## 6 Experiments

In this section, we investigate three primary questions. First, in the tabular setting, can our algorithm InAC converge to a policy found by an oracle method that exactly eliminates out-of-distribution (OOD) actions when bootstrapping? Second, in Mujoco benchmarks, how does our algorithm compare with several baselines using different offline datasets with different coverage? Third, how does InAC compare with other baselines when used for online fine-tuning after offline training? We refer readers to Appendix B for additional details and supplementary experiments.

**Baseline algorithms:** Oracle-Max: completely eliminates OOD actions when bootstrapping in tabular domains, by using counts to exactly estimate $\pi_\mathcal{D}$. FQI: the regular Q-learning update applied to batch offline data. CQL (Kumar et al., 2020): conservative Q-learning. IQL (Kostrikov et al., 2022): implicit Q-learning. TD3+BC (Fujimoto & Gu, 2021): TD3 with behavior cloning regularization. AWAC (Nair et al., 2021): Advantage Weighted Actor-Critic.

### 6.1 Sanity Check: approaching oracle performance in the Tabular Setting

In this experiment we demonstrate that InAC finds the same policy as found by an oracle algorithm that completely removes out-of-distribution (OOD) actions. We use the Four Rooms environment, where the agent starts from the bottom-left and needs to navigate through the four rooms to reach the goal in the up-right corner in as few steps as possible. There are four actions: $\mathcal{A} = \{up, down, right, left\}$. The reward is zero on each time step until the agent reaches the goal-state where it receives +1. Episodes are terminated after 100 steps, and $\gamma$ is 0.9. We use three different behavior policies to collect three datasets from this environment called **Expert, Random, and Missing-Action**. The Expert dataset contains data collected by the optimal policy. In Random dataset, the behavior policy takes each action with equal probability. For the Missing-Action dataset, we removed all transitions taking $down$ actions in the upper-left room from the Mixed dataset.

To magnify the impact of bootstrapping from OOD actions we used optimistic initialization for each algorithm (i.e., initialized all action values to be larger than the actual values under the optimal policy). This ensures overestimation occurs in some states and we can observe how well the algorithms mitigate poor bootstrap targets.

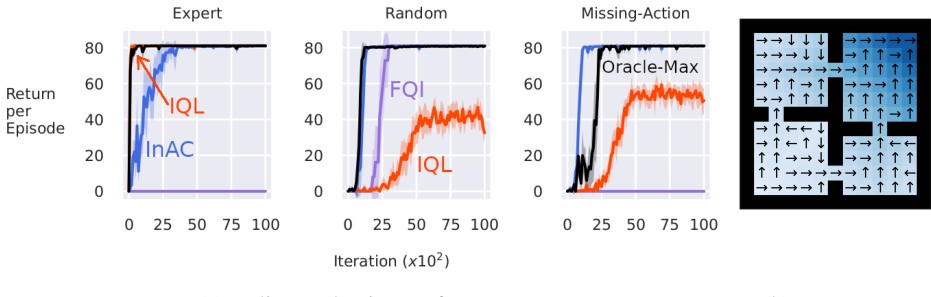

(a) Policy evaluation performance    (b) Four Room

Figure 1: Policy evaluation performance (return per episode) v.s. number of updates on Expert, Random, and Missing-Action datasets. Each curve is averaged over 10 runs, and shaded areas show a 95% confidence interval.

The results in Figure 1 are exactly as expected. InAC converges to the same policy as found by Oracle-Max. The FQI baseline cannot effectively remove OOD actions when bootstrapping, and so performs poorly and sometimes completely fails when the dataset has poor action coverage (i.e., there are many OOD actions). Finally, IQL performs poorly when the offline data is highly skewed towards suboptimal policies. It is likely because the upper expectile of the state value provides a poor approximation to the in-sample maximum action value.

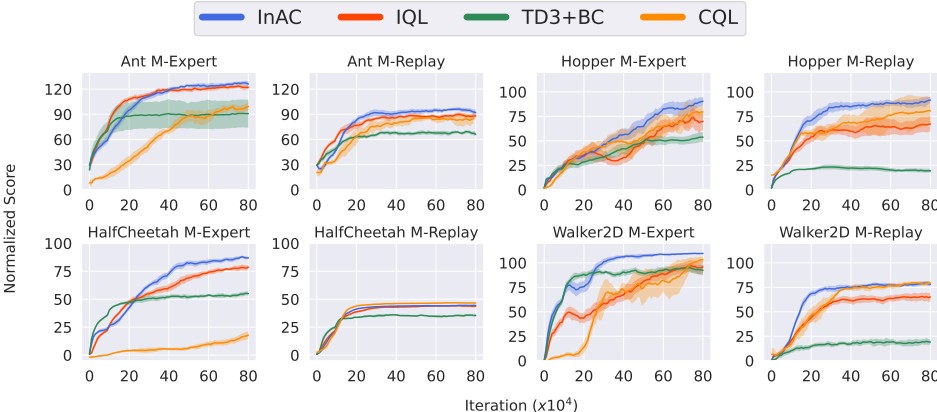

Figure 3: Policy evaluation performance (normalized score) v.s. number of updates. *M* denotes Medium. The results are averaged over 10 runs, after using a smoothing window of size 10. Results on additional offline datasets are in Appendix B.1, showing that InAC still learned the best policy.

## 6.2 OOD EFFECTS IN CONTINOUS ACTION PROBLEMS

In this section we provide a suite of results from four Mujoco environments from D4RL (Fu et al., 2020), now standard datasets for evaluating offline RL algorithms. Each dataset (named as Expert, Medium-Expert, Medium-Replay, and Medium) was designed to mimic different deployment scenarios. In the Expert dataset all trajectories were collected using a policy learned by a SAC agent. In Medium, all trajectories were collected with the policy learned by a SAC agent halfway thought training. Medium-Expert combines the expert and medium datasets together, and similarly Medium-Replay combines Medium with the replay buffer used during learning.

Figure 2 summarizes each algorithm's performance averaged over all environments under different datasets. Our algorithm's performance dominates the others across datasets. In Figure 3 we provide a more detailed view of the data with learning curves in each environment. Overall InAC performs best or nearly so across all domains. In Hopper M-Expert, the result is likely a three-way tie, while in HalfCheetah M-Expert TD3+BC learns faster initially, but the quickly converges to much lower final performance compared with InAC. Naturally, all methods are dependent on the quality of the dataset. For example, when shifting from the higher quality (medium-expert) to the lower quality (medium-replay) data, TD3+BC—which regularizes the policy to stay close to the behavior policy—exhibits a significant performance drop. Overall, TD3+BC and CQL's performance is problem dependent: in some problems performing well and in others basically failing to learn. Finally IQL performs nearly as well as InAC on many problems, but notably not on Walker2D and Hopper M-Replay.

These results provide evidence that explicitly avoiding bootstrapping from OOD actions provides a significant benefit, but that regularizing the learned policy to stay close to the behavior policy can be problematic.

## 6.3 FROM OFFLINE TRAINING TO ONLINE FINE-TUNING

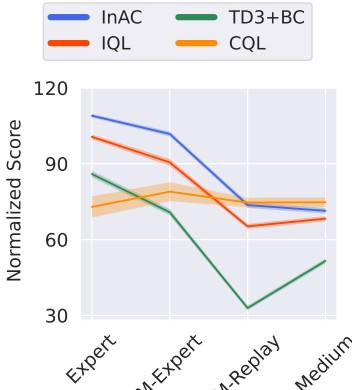

Figure 2: Averaged score over environments v.s. different offline datasets. We averaged the normalized score over four Mujoco tasks and 10 runs. The shaded area indicates the 95% confidence interval. Comparing InAC and IQL with the *sign test* over 40 runs, InAC was significantly better in all datasets. Expert, M-Expert, and M-Replay had $p$-value close to 0, while Medium dataset gave $p = 0.002$.

In real-world applications, it can be useful to take an offline-trained deep RL agent and fine-tune it online. In this section, we investigate how the performance of different baselines changes in fine-tuning. At the beginning of fine-tuning, the agent's policy is initialized with the policy learned offline and the buffer is filled with that same offline dataset. During online interactions, the agent continually adds its new experience into the buffer.

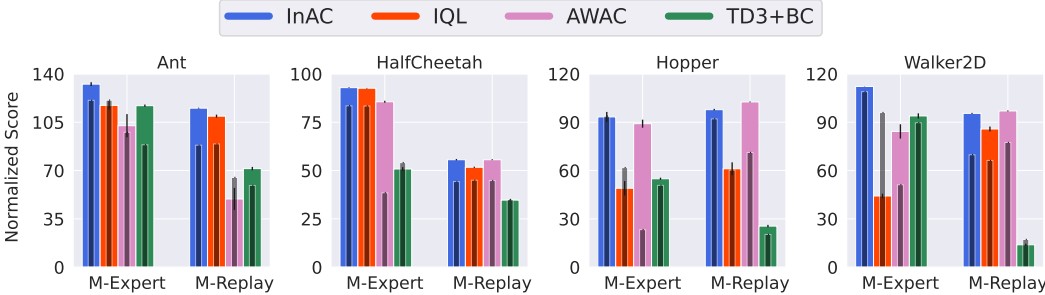

Figure 4: Online fine-tuned performance on Medium-Expert and Medium-Replay datasets across four Mujoco environments. *M* represents Medium in this Figure. The results were averaged over 10 random seeds. The short vertical line indicates the range of 3 times standard error. Each **colored bar** shows the performance after 0.8M steps of fine-tuning. The **thinner black bar inside the colored bar** indicates the performance immediately after offline training (i.e., before online fine-tuning). We also report numerical numbers in the Table 13 in Appendix B.1.

Figure 4 shows the policy performance before and after online fine-tuning. We see that InAC is consistently one of the best algorithms across these environments and datasets. There are a few particularly notable outcomes in these experiments. In Hopper and Walker2d for the Medium-Expert data, the performance for IQL drops significantly after fine-tuning. This contrasts all the other algorithms, which maintained or improved performance when fine-tuning. The cause of this drop is as yet unclear. There is one new algorithm in this set, called AWAC, which was originally proposed specifically for online fine-tuning setting (Nair et al., 2021). We do in-fact see that this algorithm can have quite poor offline performance, but significantly improve after fine-tuning. Despite being designed for this fine-tuning setting, however, it does not outperform the offline algorithms, except in Hopper with Medium-Replay and more minorly on Walker2d with Medium-Replay. Overall, we find that InAC performs well in both the fully offline setting as well as when incorporating fine-tuning.

## 7 CONCLUSION

In this paper we considered the problem of learning action-values and corresponding policies from a fixed batch of data. The algorithms designed for this setting need to account for the fact that action-coverage may be partial: certain actions may never be taken in certain regions of the state space. This complicates learning with our algorithms that rely on action-values estimates $q(s, a)$ and bootstrapping. In particular, if an action $a$ is not visited in $s$ or similar states, the $q(s, a)$ can be an arbitrary value. If this arbitrary value is high, it is likely to be used in the max in the bootstrap target and used to update the policy, which increases probability for high-value actions. This agent is chasing hallucinations, that can produce poor policies or even divergence. We focused on a simple approach to mitigate this issue: redefining the objectives to use an in-sample softmax and finally obtaining an approach to update towards an in-sample soft greedy policy that only uses actions sampled from the dataset. The resulting In-sample AC algorithm avoids these hallucinated values when updating the actor, and so correspondingly avoids them when updating the values.

We had two clear findings from this work. First, the move to an in-sample softmax was a key step towards a simple implementation of in-sample learning. Previous work, like BCQ, tried to produce a simple algorithm built on an in-sample max algorithm, but needed to incorporate several tricks and later algorithms significantly improve on it. In-sample AC, on the other hand, required only minor modifications to existing AC approaches. The actor update was modified to consider the in-sample softmax, but the resulting update was no more complex than typical actor updates. Second, our results indicate that overall Implicit Q-learning (IQL) is quite a good algorithm. Like In-sample AC, it also avoids relying on actions sampled from an approximation of $\pi_{\mathcal{D}}$, but does so using expectile regression. Nonetheless, we find that In-sample AC is always competitive with IQL, and in some cases significantly outperforms it when the dataset is generated by a more suboptimal behavior policy. IQL can still be skewed by too many suboptimal actions in the dataset. In-sample AC provides a simple, easy-to-use approach, for both discrete and continuous actions, with an update designed to match only the support of $\pi_{\mathcal{D}}$ and not the action probabilities.

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

## A   Appendix: Proofs

This section includes the proof of all main results.

### A.1   Results for one-step decision making

We first introduce some results for one-step decision making that will be used in the derivations of main results.

**Maximum Entropy Optimization**   We consider a $k$-armed one-step decision making problem. Let $\Delta$ be a $k$-dimensional simplex and $\boldsymbol{q} = (q(1), \ldots, q(k)) \in \mathbb{R}^k$ be the reward vector. Maximum entropy optimization considers

$$\max_{\pi \in \Delta} \ \pi \cdot \boldsymbol{q} + \tau \mathbb{H}(\pi) \,. \tag{17}$$

The next result characterizes the solution of this problem (Lemma 4 of (Nachum et al., 2017)).

**Lemma 2.** *For $\tau > 0$, let*

$$F_\tau(\boldsymbol{q}) = \tau \log \sum_a e^{q(a)/\tau} \,, \quad f_\tau(\boldsymbol{q}) = \frac{e^{\boldsymbol{q}/\tau}}{\sum_a e^{q(a)/\tau}} = e^{\frac{\boldsymbol{q} - F_\tau(\boldsymbol{q})}{\tau}} \,. \tag{18}$$

*Then there is*

$$F_\tau(\boldsymbol{q}) = \max_{\pi \in \Delta} \ \pi \cdot \boldsymbol{q} + \tau \mathbb{H}(\pi) = f_\tau(\boldsymbol{q}) \cdot \boldsymbol{q} + \tau \mathbb{H}(f_\tau(\boldsymbol{q})) \,. \tag{19}$$

**In-Sample Maximum Entropy Optimization**   Let $\beta \in \Delta$ be an arbitrary policy. In-sample maximum entropy optimization considers

$$\max_{\pi \preceq \beta} \ \pi \cdot \boldsymbol{q} + \tau \mathbb{H}(\pi) \,. \tag{20}$$

We now characterize the solution of this problem. For $\tau > 0$ define the *in-sample softmax value*,

$$F_{\beta,\tau}(\boldsymbol{q}) = \tau \log \left( \sum_{a:\beta(a)>0} e^{q(a)/\tau} \right) , \tag{21}$$

and the *in-sample softmax policy*,

$$f_{\beta,\tau}(\boldsymbol{q}) = \frac{\beta e^{\boldsymbol{q}/\tau - \log \beta}}{\sum_{a:\beta(a)>0} e^{q(a)/\tau}} = \beta e^{\frac{\boldsymbol{q} - F_{\beta,\tau}(\boldsymbol{q})}{\tau} - \log \beta} \,. \tag{22}$$

**Lemma 3.**

$$F_{\beta,\tau}(\boldsymbol{q}) = \max_{\pi \preceq \beta} \ \pi \cdot \boldsymbol{q} + \tau \mathbb{H}(\pi) = f_{\beta,\tau}(\boldsymbol{q}) \cdot \boldsymbol{q} + \tau \mathbb{H}(f_{\beta,\tau}(\boldsymbol{q})) \,. \tag{23}$$

*Proof.* This result is directly implied by Lemma 2. $\qquad\square$

The next result shows that $F_{\beta,\tau}$ is a contractor.

**Lemma 4.** *For any two vectors $\boldsymbol{q}_1, \boldsymbol{q}_2 \in \mathbb{R}^k$,*

$$|F_{\beta,\tau}(\boldsymbol{q}_1) - F_{\beta,\tau}(\boldsymbol{q}_2)| \le \|\boldsymbol{q}_1 - \boldsymbol{q}_2\|_\infty \,. \tag{24}$$

*Proof.*

$$F_{\beta,\tau}(\boldsymbol{q}_1) - F_{\beta,\tau}(\boldsymbol{q}_2) = \sup_{\pi_1 \preceq \beta} \{\pi_1 \cdot \boldsymbol{q}_1 + \tau \mathbb{H}(\pi_1)\} - \sup_{\pi_2 \preceq \beta} \{\pi_2 \cdot \boldsymbol{q}_2 + \tau \mathbb{H}(\pi_2)\} \tag{25}$$

$$= \sup_{\pi_1 \preceq \beta} \left\{ \inf_{\pi_2 \preceq \beta} \pi_1 \cdot \boldsymbol{q}_1 - \pi_2 \cdot \boldsymbol{q}_2 + \tau \mathbb{H}(\pi_1) - \tau \mathbb{H}(\pi_2) \right\} \tag{26}$$

$$\leq \sup_{\pi \preceq \beta} \{\pi \cdot \boldsymbol{q}_1 - \pi \cdot \boldsymbol{q}_2\} \tag{27}$$

$$\leq \max_{a:\beta(a)>0} q_1(a) - q_2(a) \tag{28}$$

$$\leq \max_a q_1(a) - q_2(a), \tag{29}$$

where the first step follows by Lemma 3, the third step follows by choosing $\pi_2 = \pi_1$. This finishes the proof.

$\square$

## A.2 RESULT FOR ON-POLICY ENTROPY-REGULARIZED BACKUP

In this section we show some basic results for on-policy entropy-regularized backup. We note that most results are generalized from Section C.2 of (Nachum et al., 2017) (which states for $\tilde{v}$) to $\tilde{q}$.

Recall that the entropy-regularized value functions are defined as

$$\tilde{q}^\pi(s,a) = r(s,a) + \gamma \mathbb{E}_{s'}[\tilde{v}^\pi(s')], \quad \tilde{v}^\pi(s) = \mathbb{E}^\pi \left[ \sum_{t=0}^\infty \gamma^t(r(s_t, a_t) - \tau \log \pi(a_t|s_t)) \Big| s_0 = s \right]. \tag{30}$$

Define the *on-policy entropy-regularized Bellman operator*

$$(\mathcal{T}^\pi q)(s,a) = r(s,a) + \gamma \mathbb{E}_{s',a'\sim P^\pi(\cdot|s,a)}[q(s',a') - \tau \log \pi(a'|s')]. \tag{31}$$

**Lemma 5.** *For any policy $\pi$, $\tilde{q}^\pi$ satisfies that $\tilde{q}^\pi = \mathcal{T}^\pi \tilde{q}^\pi$. Moreover, suppose that $|\mathcal{A}| < \infty$, $\mathcal{T}^\pi$ is a contraction mapping.*

*Proof.* By the definition of $\tilde{v}^\pi$ and $\tilde{q}^\pi$,

$$\tilde{v}^\pi(s) = \mathbb{E}^\pi \left[ \sum_{t=0}^\infty \gamma^t(r(s_t, a_t) - \tau \log \pi(a_t|s_t)) \Big| s_0 = s \right] \tag{32}$$

$$= \mathbb{E}^\pi \left[ r(s_0, a_0) - \tau \log \pi(a_0|s_0) + \gamma \sum_{i=0}^\infty \gamma^i(r(s_{i+1}, a_{i+1}) - \tau \log \pi(a_{i+1}|s_{i+1})) \Big| s_0 = s \right] \tag{33}$$

$$= \mathbb{E}_{a\sim\pi(s)} \left[ r(s,a) - \tau \log \pi(a|s) + \gamma \mathbb{E}_{s'} \left[ \mathbb{E}^\pi \left[ \sum_{t=0}^\infty \gamma^t(r(s_t, a_t) - \tau \log \pi(a_t|s_t)) \Big| s_0 = s' \right] \right] \right] \tag{34}$$

$$= \mathbb{E}_{a\sim\pi(s)} \left[ r(s,a) - \tau \log \pi(a|s) + \gamma \mathbb{E}_{s'\sim P(s,a)}[\tilde{v}^\pi(s')] \right] \tag{35}$$

$$= \mathbb{E}_{a\sim\pi(s)} [\tilde{q}^\pi(s,a) - \tau \log \pi(a|s)]. \tag{36}$$

Thus

$$\tilde{q}^\pi(s,a) = r(s,a) + \mathbb{E}_{s'}[\tilde{v}^\pi(s')] \tag{37}$$

$$= r(s,a) + \mathbb{E}_{s'}[\mathbb{E}_{a\sim\pi(s')}[\tilde{q}^\pi(s',a') - \tau \log \pi(a'|s')]] \tag{38}$$

$$= r(s,a) + \mathbb{E}_{s',a'\sim P^\pi(s,a)}[\tilde{q}^\pi(s',a') - \tau \log \pi(a'|s')] \tag{39}$$

$$= \mathcal{T}^\pi \tilde{q}^\pi. \tag{40}$$

This finishes the proof of the first part. Since $|\mathcal{A}| < \infty$, $\log \pi(a|s)$ is bounded for any $s, a$. Then that $\mathcal{T}^\pi$ is a contraction mapping follows directly from standard argument (Puterman, 2014). $\square$

This shows that $\tilde{q}^\pi$ is a fixed of $\mathcal{T}^\pi$. That is, starting from any value $q$, we can learn $\tilde{q}^\pi$ by repeatedly applying $q = \mathcal{T}^\pi q$. The next result characterizes the convergence rate of this algorithm.

**Lemma 6.** *For any $\pi$ and $q$, we have*

$$\|(\mathcal{T}^\pi)^k q - \tilde{q}^\pi\|_\infty \leq \gamma^k \|q - \tilde{q}^\pi\|_\infty . \tag{41}$$

*Proof.* We prove the result by induction. For the base case, $k = 0$, the result trivially follows. Now suppose that the result holds for $k - 1$. Then

$$\|(\mathcal{T}^\pi)^k q - \tilde{q}^\pi\|_\infty = \max_{s,a} \left|(\mathcal{T}^\pi)^k q(s, a) - \tilde{q}^\pi(s, a)\right| \tag{42}$$

$$= \max_{s,a} \left|\mathcal{T}^\pi(\mathcal{T}^\pi)^{k-1} q(s, a) - \mathcal{T}^\pi \tilde{q}^\pi(s, a)\right| \tag{43}$$

$$= \gamma \max_{s,a} \left|\mathbb{E}_{s',a' \sim P^\pi(s,a)} \left[(\mathcal{T}^\pi)^{k-1} q(s', a') - \tilde{q}^\pi(s', a')\right]\right| \tag{44}$$

$$\leq \gamma \max_{s,a} \left|(\mathcal{T}^\pi)^{k-1} q(s, a) - \tilde{q}^\pi(s, a)\right| \tag{45}$$

$$= \gamma^k \|q - \tilde{q}^\pi\|_\infty , \tag{46}$$

where the second step uses Lemma 5, the third step uses the definition of $\mathcal{T}^\pi$, the fourth step uses the Holder's inequality, the last step uses the induction hypothesis. This finishes the proof. $\square$

Finally, we also need the monotonicity property of the on-policy Bellman operation.

**Lemma 7.** *For any $\pi$, if $q_1 \geq q_2$, then $\mathcal{T}^\pi q_1 \geq \mathcal{T}^\pi q_2$.*

*Proof.* Assume $q_1 \geq q_2$ and note that for any state-action $s, a$

$$(\mathcal{T}^\pi q_1)(s, a) - (\mathcal{T}^\pi q_2)(s, a) = \gamma \mathbb{E}_{s',a' \sim P^\pi(s,a)} \left[q_1(s', a') - q_2(s', a')\right] \geq 0 . \tag{47}$$

$\square$

Policy improvement lemma.

**Lemma 8.** *Let $\pi$ be a policy such that $\pi \preceq \beta$. Define $\pi'$*

$$\pi'(\cdot|s) \propto \beta(\cdot|s) \exp\left(\frac{\tilde{q}^\pi(s, :)}{\tau} - \log \beta(\cdot|s)\right) . \tag{48}$$

*Then $\pi' \preceq \beta$ and $\tilde{q}^{\pi'} \geq \tilde{q}^\pi$.*

*Proof.* The first part trivially holds by the definition of $\pi'$.

For the second part, note that by Lemma 3, for any state $s \in \mathcal{S}$,

$$\pi'(\cdot|s) \cdot (\tilde{q}^\pi(s, :) - \tau \log \tilde{\pi}(\cdot|s)) \geq \pi(\cdot|s) \cdot (\tilde{q}^\pi(s, :) - \tau \log \pi(\cdot|s)) . \tag{49}$$

Then by Lemma 5, for any $s, a \in \mathcal{S} \times \mathcal{A}$,

$$\tilde{q}^\pi(s, a) = r(s, a) + \gamma \mathbb{E}_{s',a' \sim P^\pi(\cdot|s,a)}[\tilde{q}^\pi(s', a') - \tau \log \pi(a'|s')] \tag{50}$$

$$\leq r(s, a) + \gamma \mathbb{E}_{s',a' \sim P^{\pi'}(\cdot|s,a)}[\tilde{q}^\pi(s', a') - \tau \log \pi'(a'|s')] \tag{51}$$

$$\leq \ldots \tag{52}$$

$$\leq \tilde{q}^{\pi'}(s, a) , \tag{53}$$

where we recursively apply Lemma 5 to expand the definition of $\tilde{q}^\pi$ and apply Eq. (49).

$\square$

A.3 RESULT FOR OFF-POLICY ENTROPY-REGULARIZED BACKUP

Given an arbitrary policy $\beta$, consider the following problem

$$\max_{\pi \preceq \beta} \tilde{v}^\pi(s) \quad \text{for all } s \in \mathcal{S} \tag{54}$$

For $\tau > 0$, define the *In-sample softmax Bellman operator*

$$(\mathcal{T}_\beta^* q)(s, a) = r(s, a) + \gamma \mathbb{E}_{s' \sim P(\cdot|s,a)} \left[ \tau \log \sum\nolimits_{a':\beta(a'|s')>0} \exp\left(q(s', a')/\tau\right) \right] \tag{55}$$

$$= r(s, a) + \gamma \mathbb{E}_{s' \sim P(\cdot|s,a)} \left[ F_{\beta,\tau}(q(s', :)) \right] \tag{56}$$

**Lemma 9.** *For $\gamma < 1$, the fixed point of the in-sample softmax Bellman operator, $q^* = \mathcal{T}_\beta^* q^*$, exists and is unique.*

*Proof.* We first show that $\mathcal{T}_\beta^*$ is a contraction. Let $q_1$ and $q_2$ be two value functions. Then

$$\left\| \mathcal{T}_\beta^* q_1 - \mathcal{T}_\beta^* q_2 \right\|_\infty = \gamma \max_{s,a} \left| \mathcal{T}_\beta^* q_1(s, a) - \mathcal{T}_\beta^* q_2(s, a) \right| \tag{57}$$

$$= \gamma \max_{s,a} \left| \mathbb{E}_{s' \sim P(\cdot|s,a)} \left[ F_{\beta,\tau}(q_1(s', :)) - F_{\beta,\tau}(q_2(s', :)) \right] \right| \tag{58}$$

$$\leq \gamma \max_s \left| F_{\beta,\tau}(q_1(s, :)) - F_{\beta,\tau}(q_2(s, :)) \right| \tag{59}$$

$$\leq \gamma \max_{s,a} |q_1(s, a) - q_2(s, a)| \tag{60}$$

$$= \gamma \| q_1 - q_2 \| , \tag{61}$$

where the second step uses the definition of $\mathcal{T}_\beta^*$, the third step uses Holder's inequality, the fourth step uses Lemma 4.

$\square$

Note that by definition, $\tilde{q}_\beta^*$ is the fixed point of $\mathcal{T}_\beta^*$.

**Lemma 10.** *If $q$ is bounded and $q \geq \mathcal{T}_\beta^* q$, then for any $\pi$, $q \geq \tilde{q}^\pi$.*

*Proof.* We first prove that for any $\pi$, $q \geq \mathcal{T}_\beta^* q$ implies that $q \geq (\mathcal{T}^\pi)^k q$ for $k \geq 0$. Then the result follows by applying Lemma 6. According to the assumption,

$$q \geq \mathcal{T}_\beta^* q = r(s, a) + \gamma \mathbb{E}_{s'} [F_{\beta,\tau}(q(s', :))] \tag{62}$$

$$\geq r(s, a) + \gamma \mathbb{E}_{s'} \left[ \sum_{a'} \pi(a'|s')(q(s', a') - \tau \log \pi(a'|s')) \right] \tag{63}$$

$$= \mathcal{T}^\pi q , \tag{64}$$

where the second inequality follows by Lemma 3. Then by Lemma 7,

$$q \geq \mathcal{T}^\pi q \geq \mathcal{T}^\pi \mathcal{T}_\beta^* q \geq \mathcal{T}^\pi \mathcal{T}^\pi q \geq \cdots \geq (\mathcal{T}^\pi)^k q . \tag{65}$$

This finishes the proof.

$\square$

We have the following key result.

**Lemma 11.** *For any $s \in \mathcal{S}$, $\tilde{v}_\beta^*(s) = \max_{\pi \preceq \beta} \tilde{v}^*(s)$.*

*Proof.* We first show $\tilde{v}_\beta^* \geq \max_{\pi \preceq \beta} \tilde{v}^\pi$. Using the definitions,

$$\tilde{v}_\beta^* = F_{\beta,\tau}(\tilde{q}_\beta^*) \tag{66}$$

$$= \tilde{\pi}_\beta^* \cdot \left( \tilde{q}_\beta^* - \tau \log \tilde{\pi}_\beta^* \right) \tag{67}$$

$$\geq \pi \cdot \left( \tilde{q}_\beta^* - \tau \log \pi \right) \quad (\pi \preceq \beta) \tag{68}$$

$$\geq \pi \cdot \left( \tilde{q}^\pi - \tau \log \pi \right) \quad (\pi \preceq \beta) \tag{69}$$

$$= \tilde{v}^\pi , \tag{70}$$

where the second and third steps follow by Lemma 3, the fourth step follows by Lemma 10, the last step follows by the definition.

We then prove $\max_{\pi \preceq \beta} \tilde{v}^\pi \geq \tilde{v}_\beta^*$ by first showing that $\tilde{q}_\beta^* = \tilde{q}^{\tilde{\pi}_\beta^*}$. Since $\tilde{q}_\beta^*$ is the fixed point $\mathcal{T}_\beta^*$, by the uniqueness of the fixed point (Lemma 9), we only need to show that $\mathcal{T}_\beta^* \tilde{q}^{\tilde{\pi}_\beta^*} = \tilde{q}^{\tilde{\pi}_\beta^*}$. This holds because for any $(s, a)$,

$$\mathcal{T}_\beta^* \tilde{q}^{\tilde{\pi}_\beta^*}(s, a) = r(s, a) + \gamma \mathbb{E}_{s'} \left[ F_{\beta, \tau}(\tilde{q}^{\tilde{\pi}_\beta^*}(s', :)) \right] \tag{71}$$

$$= r(s, a) + \gamma \mathbb{E}_{s'} \left[ \sum_{a'} \tilde{\pi}_\beta^*(a'|s') \left( \tilde{q}^{\tilde{\pi}_\beta^*}(s', a') - \tau \log \tilde{\pi}_\beta^*(a'|s') \right) \right] \tag{72}$$

$$= \mathcal{T}^{\tilde{\pi}_\beta^*} \tilde{q}^{\tilde{\pi}_\beta^*}(s, a) \tag{73}$$

$$= \tilde{q}^{\tilde{\pi}_\beta^*}(s, a), \tag{74}$$

where the second step uses Lemma 3 and the last step uses Lemma 5. Then,

$$\max_{\pi \preceq \beta} \tilde{v}^\pi \geq \tilde{v}^{\tilde{\pi}_\beta^*} = \tilde{\pi}_\beta^* \cdot \left( \tilde{q}^{\tilde{\pi}_\beta^*} - \tau \log \tilde{\pi}_\beta^* \right) = \tilde{\pi}_\beta^* \cdot \left( \tilde{q}_\beta^* - \tau \log \tilde{\pi}_\beta^* \right) = \tilde{v}_\beta^*, \tag{75}$$

where the second equality uses that $\tilde{q}_\beta^* = \tilde{q}^{\tilde{\pi}_\beta^*}$, the last step uses Lemma 3. This finishes the proof.

$\square$

## A.4 PROOF OF THEOREM 1

**Theorem 4** (Restatement of Theorem 1). *Let $\tilde{q}_\beta^*$ be a value function recursively defined as*

$$\tilde{q}_\beta^*(s, a) = r(s, a) + \gamma \mathbb{E}_{s' \sim P(\cdot|s,a)} \left[ \tau \log \sum_{a':\beta(a'|s')>0} \exp\left( \tilde{q}_\beta^*(s', a')/\tau \right) \right], \tag{76}$$

*and $\tilde{\pi}_\beta^*$ be a policy defined as*

$$\tilde{\pi}_\beta^*(a|s) \propto \beta(a|s) \exp\left( \tilde{q}_\beta^*(s, a)/\tau - \log \beta(a|s) \right). \tag{77}$$

*Then we have $\tilde{q}_\beta^* \to q_\beta^*$ and $\tilde{\pi}_\beta^* \to \pi_\beta^*$ as $\tau \to 0$.*

*Proof.* By Lemma 11, we have $\tilde{q}_\beta^*(s, a) = \max_{\pi \preceq \beta} \tilde{q}^*(s, a)$ for any $s, a$. The result directly follows by definition of $\tilde{q}^*$. The result for policy can be proved similarly.

$\square$

## B APPENDIX FOR EXPERIMENTS

### B.1 ADDITIONAL EXPERIMENTS

This section includes additional experiments to investigate the following questions.

1. We used optimistic initialization for all algorithms in the tabular domain. How do the algorithms perform when using a zero/pessimistic initialization? We show this in Figure 5. In the meanwhile, we added Mixed dataset, which has $1\%$ optimal trajectories and $99\%$ random trajectories.

2. How do our algorithms work on the discrete action domains in the deep learning setting? We show the learning curves on Mountain Car, Lunar Lander, Acrobot in Figure 7. The final performance is listed in Figure 8 with a normalized score, while the absolute score can be found in Figure 9.

3. How do our algorithms work on more datasets in the continuous action domains? We put learning curves for the expert and medium dataset in Figure 10, then list the performance of policies learned with all baselines and all datasets in Figure 11 with a normalized score,

while the absolute score can be found in Figure 12. For more Fine-tuning results, we put them in Figure 13 with a normalized score, while the absolute score can be found in Figure 14.

4. Will longer run reduce the gap between InAC and baselines? Can InAC still learn better or similar policy compared to baselines, if we use another common batch size setting in Mujoco tasks (256)? We changed the batch size to 256 and increased the number of iteration to 1.2 million, then show the performance in Figure 15.

5. How does InAC perform in AntMaze? We used antmaze-umaze-v0 and antmaze-umaze-diverse-v0 to test InAC, then added the learning curve comparing InAC to IQL in Figure 16. We followed the set up in previous work (Kostrikov et al., 2022).

## B.2 REPRODUCING DETAILS

This section includes all experimental details to reproduce any empirical results in this paper. We use python version 3.9.6, gym version 0.10.0, pytorch version 1.10.0.

## B.3 REPRODUCING DETAILS ON TABULAR DOMAIN

**Four room environment:** The environment is a $13 \times 13$ gridword, with walls separating the whole space into 4 rooms (as shown in Figure 6). The black area refers to the wall. The agent starts from the lower-left corner and learns to stay in the upper-right corner. When an agent runs into the wall, it returns to its previous state. The agent gets a $+1$ when it transits to the state in the upper-right corner and gets 0 otherwise. The discount rate is 0.9. Thus the upper bound of state value is 10. In tabular experiments, each trajectory was limited to 100 steps. $\tau$ was set to 0.01. A mini-batch update was used. The agent sampled 100 transitions at each iteration. Among the 4 datasets we used, mixed and random datasets used the random restart to ensure full state-action pairs coverage, while expert and missing-action datasets did not.

**Offline data collection:** We used value iterations to find the optimal policy (10k iterations). For the expert dataset, we collect 10k transitions with the optimal policy. For the random dataset, we collected 10k transitions with a random restart and equal probability of taking each action. The mixed dataset consists of 100 transitions from the expert dataset and 9900 transitions from the random dataset. The missing action dataset is constructed by all going-down transitions in the upper-left room from the mixed dataset.

**Algorithm parameter sweep:** The learning rate of InAC, Oracle-Max, and FQI was swept in $[0.1, 0.03, 0.01, 0.003, 0.001]$. Sarsa had a larger range: $[0.1, 0.03, 0.01, 0.003, 0.001, 0.0003, 0.0001, 0.00003]$. The $\tau$ of in-sample methods was fixed to 0.01.

## B.4 REPRODUCING DETAILS OF DEEP RL ALGORITHMS

**Network architecture:** In mujoco tasks, we used 2 hidden layers with 256 nodes each for all neural networks. In discrete action environments, we used 2 hidden layers with 64 nodes each.

**Offline data generation details:** In continuous control tasks, we used the datasets provided by D4RL. In discrete control tasks, we used a well-trained DQN agent to collect data. The DQN agent had 2 hidden layers with 64 nodes on each, with FTA (Pan et al., 2021) activation function on the last hidden layer and ReLU on others. In Acrobot, the agent was trained for 40k steps with batch size 64. In Lunar Lander and Mountain Car, we trained the agent for 500k and 60k steps separately, with other settings the same as in Acrobot. The expert dataset contains 50k transitions collected with the fixed policy learned by the DQN agent. The mixed dataset has 2k (4%) near-optimal transitions and 48k (96%) transitions collected with a randomly initialized policy.

**Offline training details:** In all tasks, we used minibatch sampling, and the mini-batch size was set to 100. We used the ADAM optimizer and ReLU activation function. The target network is updated by using Polyak average: $0.995 \times target\_weight + 0.005 \times learning\_weight$. We trained the agent for 0.8 million iterations and 70k iterations in mujoco and discrete action environments respectively.

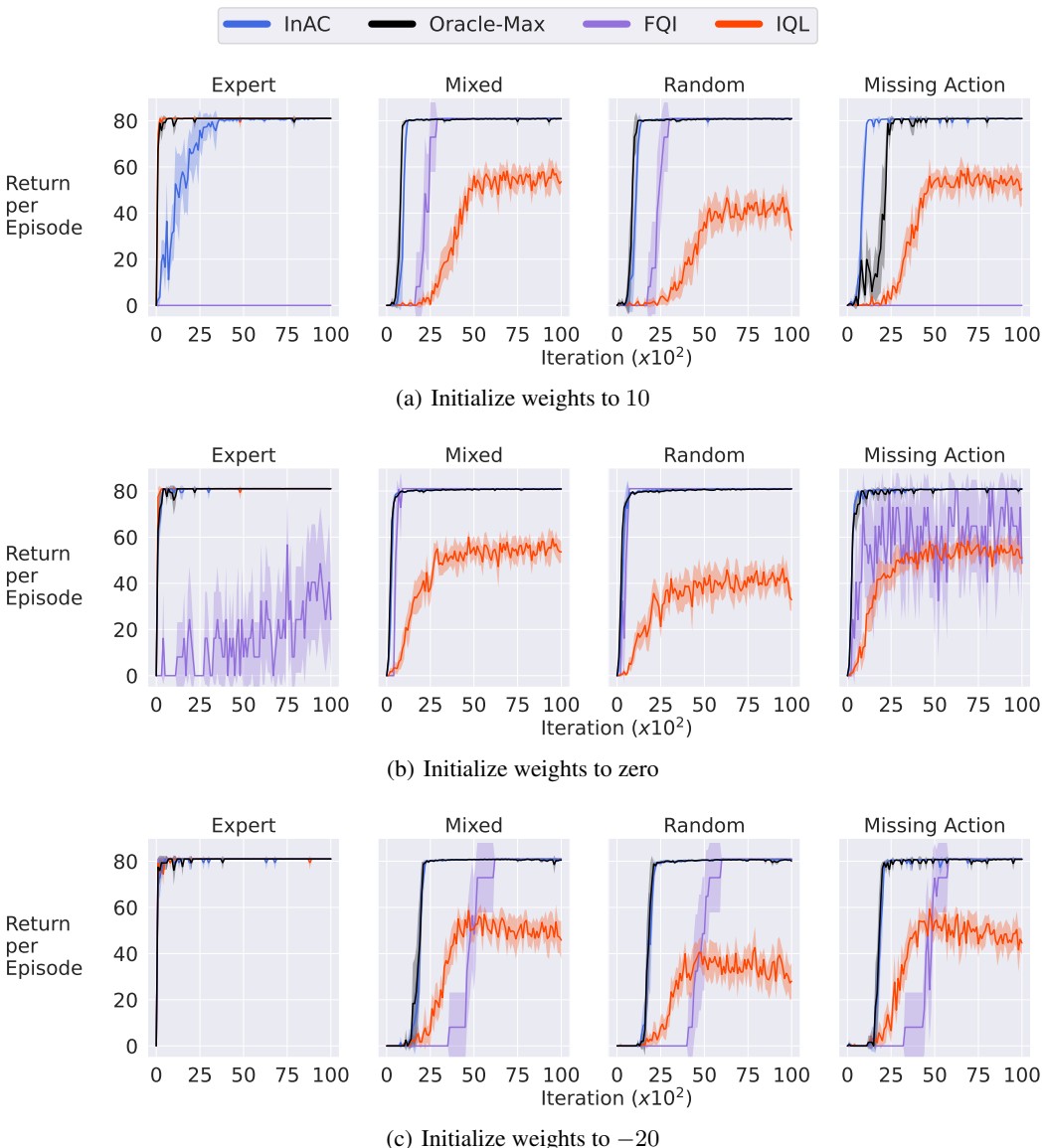

Figure 5: Learning curves under different initialization on our four room gridworld tabular domain. (a) used 10 to initialize weights, (b) used 0, and (c) used −20. InAC learns reasonable policy with 1) expert trajectories, 2) missing action trajectories, 3) mixed trajectories, and 4) random trajectories, where 3) and 4) have full state-action coverage. The results were averaged over 10 random seeds, except that CQL had 5 seeds. The shaded area indicates 95% confidence interval.

**Fine Tuning details:** We kept all settings as same as in offline learning, and used the learned policy as initialization. The offline data was filled into the buffer at the beginning of fine-tuning. New interactions would be appended to the buffer later in fine-tuning. No data were removed. The fine-tuning had 0.8M steps.

**Policy evaluation details:** The policy was evaluated for 5 episodes in the true environment with a timeout setting. Acrobot and Lunar Lander had timeout=500, Mountain Car used 2000, and mujoco tasks used 1000. The numbers reported were averaged over 10 random seeds.

**Algorithm parameter setting:**

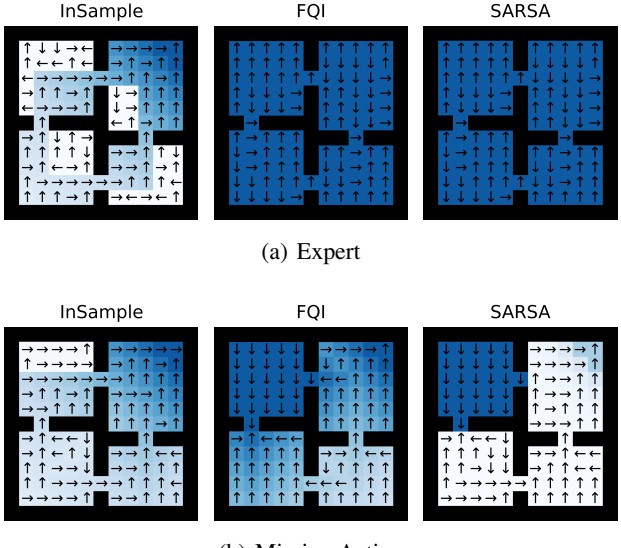

Figure 6: Visualization of the learned policy of each algorithms and the estimated values at each state. The blue colors indicate the action value of the corresponding policy and the arrow indicates the action taken by the learned policy. A deeper color refers to a higher action value. It is clear that both FQI and SARSA have serious overestimation and found an incorrect policy when the offline data lacks action coverage (i.e., on the Expert and Missing-action offline data).

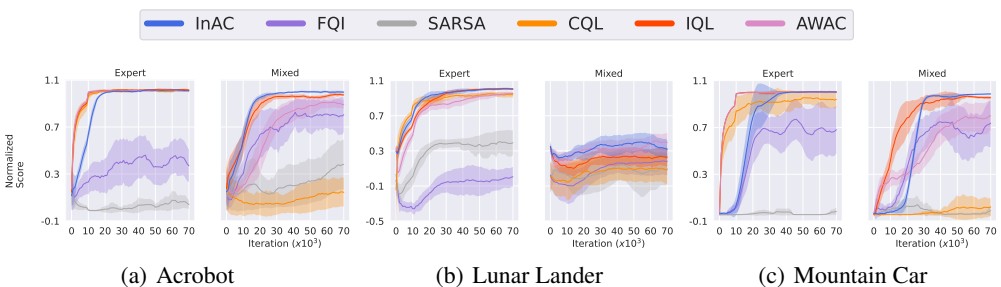

Figure 7: Learning curves on the discrete action domains. InAC learns the best policy or a comparable policy to the strongest baseline. From left to right, we show the result in Acrobot, Lunar Lander, and Mountain Car. In each environment, we tested 2 datasets: Expert and Mixed. In expert dataset, all trajectories were collected with the near-optimal policy, while in mixed dataset, $4\%$ trajectories were optimal and $96\%$ were collected with a randomly initialized neural network policy. The y-axis is a normalized return reflecting the performance. The return was normalized according to returns obtained by a well trained DQN agent (upper bound) and a randomly initialized network (lower bound). The higher the normalized value, the better the performance. The curves were smoothed with window length10. The results were averaged over 10 random seeds. The shaded area indicates $95\%$ confidence interval.

Mujoco tasks: For all algorithms, the learning rate was swept in $\{3 \times 10^{-4}, 1 \times 10^{-4}, 3 \times 10^{-5}\}$. InAC swept $\tau$ in $\{1.0, 0.5, 0.33, 0.1, 0.01\}$. AWAC swept $\lambda$ in $\{1.0, 0.5, 0.33, 0.1, 0.01\}$. IQL swept expectile in $\{0.9, 0.7\}$ and temperature in $\{10.0, 3.0\}$. The number came from what was reported in the original IQL paper. TD3+BC used $\alpha = 2.5$ as in the original paper. CQL-SAC used automatic entropy tuning as in the original paper.

Discrete action environments: For all algorithms, the learning rate was swept in $\{0.003, 0.001, 0.0003, 0.0001, 3e-5, 1e-5\}$. For InAC, $\tau$ was swept in $\{1.0, 0.5, 0.1, 0.05, 0.01\}$. IQL had the same parameter sweeping range as in mujoco tasks. AWAC used $\lambda = 1.0$ as in the original paper. CQL used $\alpha = 5.0$.

| Environment | Dataset | InAC | FQI | SARSA | CQL | IQL | AWAC | Oracle-Max |
|---|---|---|---|---|---|---|---|---|
| Acrobot | opt | 1.0116 (0.0006) | 0.3373 (0.0162) | 0.0597 (0.0061) | 1.0040 (0.0011) | 1.0047 (0.0015) | 1.0069 (0.0007) | **1.0214 (0.0007)** |
| | mixed | **0.9970 (0.0009)** | 0.8851 (0.0036) | 0.4041 (0.0153) | 0.0934 (0.0084) | 0.9861 (0.0012) | 0.8796 (0.0041) | 0.8589 (0.0074) |
| Lunar Lander | opt | 0.9956 (0.0014) | -0.0176 (0.0105) | 0.3872 (0.0088) | 0.9229 (0.0031) | **1.0108 (0.0008)** | 0.9648 (0.0017) | 0.9424 (0.0035) |
| | mixed | 0.2960 (0.0057) | 0.2281 (0.0099) | 0.1087 (0.0109) | 0.0783 (0.0105) | 0.1832 (0.0095) | **0.3632 (0.0094)** | 0.3064 (0.0103) |
| Mountain Car | opt | 1.0017 (0.0004) | 0.7802 (0.0124) | -0.0254 (0.0021) | 0.9423 (0.0050) | 1.0037 (0.0004) | **1.0061 (0.0001)** | 0.9973 (0.0006) |
| | mixed | **0.9833 (0.0006)** | 0.7226 (0.0152) | 0.0130 (0.0065) | 0.0330 (0.0066) | 0.9405 (0.0015) | 0.8182 (0.0073) | 0.9628 (0.0016) |

Figure 8: The offline-trained final performance of each algorithm in discrete action space environments. The number in bracket is the standard error. Scores are normalized. The bold numbers are the best performance in the same setting. Performance was averaged over 10 random seeds.

| Environment | Dataset | InAC | FQI | SARSA | CQL | IQL | AWAC | Oracle-Max |
|---|---|---|---|---|---|---|---|---|
| Acrobot | opt | -85.14 (0.22) | -357.48 (6.56) | -469.58 (2.45) | -88.18 (0.44) | -87.92 (0.60) | -87.04 (0.29) | **-81.18 (0.28)** |
| | mixed | **-91.02 (0.38)** | -136.22 (1.45) | -330.48 (6.20) | -455.96 (3.41) | -95.44 (0.50) | -138.44 (1.66) | -146.80 (2.99) |
| Lunar Lander | opt | 201.20 (0.90) | -460.09 (6.85) | -195.86 (5.76) | 153.71 (2.05) | **211.09 (0.50)** | 181.06 (1.11) | 166.49 (2.29) |
| | mixed | -255.43 (3.73) | -299.73 (6.48) | -377.64 (7.11) | -397.51 (6.85) | -328.99 (6.18) | **-211.53 (6.13)** | -248.59 (6.70) |
| Mountain Car | opt | -118.14 (0.66) | -516.46 (22.35) | -1965.74 (3.86) | -224.92 (8.95) | -114.46 (0.65) | **-110.08 (0.22)** | -125.90 (1.12) |
| | mixed | **-151.18 (1.11)** | -620.12 (27.34) | -1896.60 (11.64) | -1860.62 (11.84) | -228.16 (2.68) | -448.14 (13.10) | -187.94 (2.88) |

Figure 9: The offline-trained final performance of each algorithm in discrete action space environments. This table reports the return per episode before normalization. The number in bracket is the standard error. The bold numbers are the best performance in the same setting. Performance was averaged over 10 random seeds.

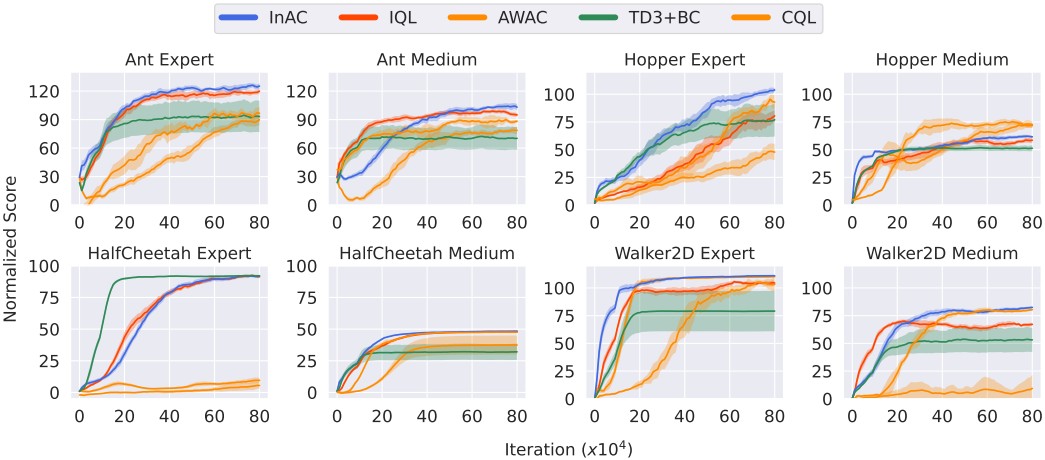

Figure 10: Learning curve on the mujoco tasks. InAC learns the best policy or a comparable policy to the strongest baseline. The results were averaged over 10 random seeds, except that CQL had 5 seeds. The shaded area indicates 95% confidence interval.

| Environment | Dataset | InAC | IQL | AWAC | TD3+BC | CQL |
|---|---|---|---|---|---|---|
| Ant | expert | **128.40 (0.42)** | 118.80 (0.50) | 100.60 (0.89) | 93.30 (1.04) | 86.00 (1.50) |
| | medium expert | 120.90 (0.58) | **121.00 (0.58)** | 97.80 (1.55) | 88.70 (0.89) | 111.40 (0.88) |
| | medium replay | 88.40 (0.63) | **89.30 (0.44)** | 65.00 (1.15) | 59.20 (0.39) | 85.00 (0.84) |
| | medium | 94.20 (0.88) | 94.50 (0.49) | 84.10 (0.30) | 69.80 (0.71) | **99.20 (0.10)** |
| HalfCheetah | expert | **93.60 (0.04)** | 91.50 (0.20) | 10.00 (0.10) | 92.10 (0.03) | 5.20 (0.17) |
| | medium expert | **83.50 (0.34)** | 83.40 (0.46) | 38.50 (0.30) | 54.10 (0.38) | 22.20 (0.12) |
| | medium replay | 44.30 (0.02) | 45.00 (0.03) | 44.80 (0.01) | 35.00 (0.20) | **46.60 (0.02)** |
| | medium | 48.30 (0.02) | **48.50 (0.02)** | 37.40 (0.43) | 32.10 (0.36) | 47.60 (0.02) |
| Hopper | expert | **103.40 (0.38)** | 89.40 (0.74) | 45.00 (0.97) | 78.90 (0.91) | 94.40 (0.61) |
| | medium expert | **93.80 (0.69)** | 61.80 (0.95) | 23.30 (0.34) | 50.80 (0.26) | 77.60 (1.10) |
| | medium replay | **92.10 (0.38)** | 60.30 (0.49) | 71.20 (0.97) | 20.30 (0.19) | 77.20 (0.92) |
| | medium | 60.30 (0.20) | 59.00 (0.28) | **72.70 (0.45)** | 50.20 (0.13) | 69.80 (0.47) |
| Walker2D | expert | **110.60 (0.09)** | 102.90 (0.60) | 110.20 (0.03) | 79.40 (1.04) | 106.00 (0.35) |
| | medium expert | **109.00 (0.10)** | 96.00 (0.35) | 51.20 (1.32) | 89.70 (0.64) | 104.60 (0.25) |
| | medium replay | 69.80 (0.57) | 66.30 (0.53) | 77.40 (0.32) | 17.20 (0.35) | **80.40 (0.18)** |
| | medium | **82.70 (0.16)** | 71.10 (0.53) | 13.20 (0.87) | 54.10 (0.65) | 82.40 (0.06) |

Figure 11: The final performance in continuous action space environments. The number in bracket is the standard error. Scores are normalized. The bold numbers are the best performance in the same setting. Performance was averaged over 10 random seeds, except that CQL had 5 seeds.

| Environment | Dataset | InAC | IQL | AWAC | TD3+BC | CQL |
|---|---|---|---|---|---|---|
| Ant | expert | **5077.97 (17.92)** | 4665.41 (20.71) | 3911.03 (37.60) | 3593.61 (43.41) | 3288.60 (63.46) |
| | medium expert | 4750.21 (24.21) | **4763.72 (24.29)** | 3787.70 (65.10) | 3403.58 (37.71) | 4369.70 (37.01) |
| | medium replay | 3391.06 (26.64) | **3427.12 (18.50)** | 2410.17 (48.23) | 2161.59 (16.45) | 3259.72 (31.60) |
| | medium | 3637.89 (36.90) | 3642.84 (20.60) | 3211.47 (12.66) | 2606.69 (29.86) | **3846.52 (4.00)** |
| HalfCheetah | expert | **11347.06 (5.12)** | 11089.65 (24.91) | 959.41 (12.87) | 11151.42 (4.46) | 354.04 (21.25) |
| | medium expert | **10086.43 (42.11)** | 10054.46 (56.34) | 4511.98 (36.84) | 6431.29 (47.99) | 2485.94 (13.24) |
| | medium replay | 5209.87 (2.66) | 5289.54 (3.22) | 5275.87 (1.27) | 4096.27 (24.64) | **5495.89 (1.68)** |
| | medium | 5716.72 (2.29) | **5742.89 (2.02)** | 4359.90 (52.57) | 3696.49 (44.49) | 5650.65 (3.18) |
| Hopper | expert | **3346.42 (12.48)** | 2885.90 (24.07) | 1445.60 (31.53) | 2544.39 (29.50) | 3053.66 (19.96) |
| | medium expert | **3032.05 (22.29)** | 1992.49 (31.03) | 733.59 (10.99) | 1635.83 (8.51) | 2504.53 (36.45) |
| | medium replay | **2975.21 (12.04)** | 1942.17 (15.85) | 2294.38 (31.45) | 638.10 (6.33) | 2501.46 (30.08) |
| | medium | 1945.62 (6.19) | 1897.93 (9.11) | **2340.96 (14.89)** | 1609.48 (4.22) | 2252.67 (15.40) |
| Walker2D | expert | **5076.38 (4.04)** | 4725.18 (27.57) | 5061.92 (1.02) | 3644.09 (47.59) | 4868.31 (15.62) |
| | medium expert | **5006.62 (4.27)** | 4410.62 (16.43) | 2361.18 (60.66) | 4120.01 (29.44) | 4804.33 (11.52) |
| | medium replay | 3205.26 (26.23) | 3034.25 (24.45) | 3553.79 (14.45) | 789.40 (15.84) | **3698.78 (8.32)** |
| | medium | **3790.83 (7.16)** | 3269.64 (24.45) | 600.84 (39.67) | 2474.75 (29.75) | 3778.23 (2.69) |

Figure 12: The absolute final performance in continuous action space environments. This table reports the score before normalization. The number in bracket is the standard error. The bold numbers are the best performance in the same setting. Performance was averaged over 10 random seeds, except that CQL had 5 seeds.

| Environment | Dataset | Performance After | InAC | IQL | AWAC | TD3+BC |
|---|---|---|---|---|---|---|
| Ant | expert | Offline | 128.40 (0.42) | 118.80 (0.50) | 100.60 (0.89) | 93.30 (1.04) |
| | | FineTune | **134.80 (0.11)** | 112.90 (0.69) | 134.70 (0.26) | 116.30 (0.73) |
| | | Change | 6.4 | -5.9 | 34.1 | 23 |
| | medium expert | Offline | 120.90 (0.58) | 121.00 (0.58) | 97.80 (1.55) | 88.70 (0.89) |
| | | FineTune | **132.60 (0.46)** | 117.20 (1.03) | 102.50 (2.80) | 117.00 (0.27) |
| | | Change | 11.7 | -3.8 | 4.7 | 28.3 |
| | medium replay | Offline | 88.40 (0.63) | 89.30 (0.44) | 65.00 (1.15) | 59.20 (0.39) |
| | | FineTune | **115.20 (0.10)** | 109.40 (0.36) | 49.40 (2.68) | 71.40 (0.41) |
| | | Change | 26.8 | 20.1 | -15.6 | 12.2 |
| | medium | Offline | 94.20 (0.88) | 94.50 (0.49) | 84.10 (0.30) | 69.80 (0.71) |
| | | FineTune | **122.50 (0.04)** | 109.20 (0.30) | -42.20 (0.25) | 82.30 (0.53) |
| | | Change | 28.3 | 14.7 | -126.3 | 12.5 |
| HalfCheetah | expert | Offline | 93.60 (0.04) | 91.50 (0.20) | 10.00 (0.10) | 92.10 (0.03) |
| | | FineTune | 92.40 (0.08) | **94.00 (0.04)** | 54.80 (0.27) | 90.90 (0.03) |
| | | Change | -1.2 | 2.5 | 44.8 | -1.2 |
| | medium expert | Offline | 83.50 (0.34) | 83.40 (0.46) | 38.50 (0.30) | 54.10 (0.38) |
| | | FineTune | **92.90 (0.05)** | 92.60 (0.05) | 85.60 (0.17) | 50.80 (0.32) |
| | | Change | 9.4 | 9.2 | 47.1 | -3.3 |
| | medium replay | Offline | 44.30 (0.02) | 45.00 (0.03) | 44.80 (0.01) | 35.00 (0.20) |
| | | FineTune | **55.60 (0.13)** | 51.70 (0.10) | **55.60 (0.08)** | 34.70 (0.12) |
| | | Change | 11.3 | 6.7 | 10.8 | -0.3 |
| | medium | Offline | 48.30 (0.02) | 48.50 (0.02) | 37.40 (0.43) | 32.10 (0.36) |
| | | FineTune | **63.80 (0.07)** | 54.80 (0.14) | 62.20 (0.09) | 42.60 (0.02) |
| | | Change | 15.5 | 6.3 | 24.8 | 10.5 |
| Hopper | expert | Offline | 103.40 (0.38) | 89.40 (0.74) | 45.00 (0.97) | 78.90 (0.91) |
| | | FineTune | **108.70 (0.17)** | 67.40 (1.02) | 77.90 (1.17) | 91.40 (0.52) |
| | | Change | 5.3 | -22 | 32.9 | 12.5 |
| | medium expert | Offline | 93.80 (0.69) | 61.80 (0.95) | 23.30 (0.34) | 50.80 (0.26) |
| | | FineTune | **93.30 (1.03)** | 49.00 (1.48) | 89.10 (0.83) | 54.90 (0.23) |
| | | Change | -0.5 | -12.8 | 65.8 | 4.1 |
| | medium replay | Offline | 92.10 (0.38) | 60.30 (0.49) | 71.20 (0.97) | 20.30 (0.19) |
| | | FineTune | 97.80 (0.17) | 61.10 (1.33) | **102.60 (0.08)** | 25.50 (0.26) |
| | | Change | 5.7 | 0.8 | 31.4 | 5.2 |
| | medium | Offline | 60.30 (0.20) | 59.00 (0.28) | 72.70 (0.45) | 50.20 (0.13) |
| | | FineTune | 80.30 (0.35) | 62.40 (0.61) | **94.60 (0.27)** | 51.10 (0.15) |
| | | Change | 20 | 3.4 | 21.9 | 0.9 |
| Walker2D | expert | Offline | 110.60 (0.09) | 102.90 (0.60) | 110.20 (0.03) | 79.40 (1.04) |
| | | FineTune | **110.90 (0.03)** | 105.50 (0.21) | 107.40 (0.44) | 108.70 (0.03) |
| | | Change | 0.3 | 2.6 | -2.8 | 29.3 |
| | medium expert | Offline | 109.00 (0.10) | 96.00 (0.35) | 51.20 (1.32) | 89.70 (0.64) |
| | | FineTune | **112.20 (0.03)** | 44.20 (0.48) | 84.30 (1.48) | 93.90 (0.54) |
| | | Change | 3.2 | -51.8 | 33.1 | 4.2 |
| | medium replay | Offline | 69.80 (0.57) | 66.30 (0.53) | 77.40 (0.32) | 17.20 (0.35) |
| | | FineTune | 95.50 (0.10) | 85.80 (0.51) | **97.00 (0.10)** | 13.90 (0.23) |
| | | Change | 25.7 | 19.5 | 19.6 | -3.3 |
| | medium | Offline | 82.70 (0.16) | 71.10 (0.53) | 13.20 (0.87) | 54.10 (0.65) |
| | | FineTune | **89.70 (0.07)** | 79.50 (0.26) | 10.10 (0.99) | 66.40 (0.27) |
| | | Change | 7 | 8.4 | -3.1 | 12.3 |

Figure 13: The performance changes during fine-tuning. The number in bracket is the standard error. Scores are normalized. Performance was averaged over 10 random seeds.

| Environment | Dataset | Performance After | InAC | IQL | AWAC | TD3+BC |
|---|---|---|---|---|---|---|
| Ant | expert | Offline | 5077.97 (17.92) | 4665.41 (20.71) | 3911.03 (37.60) | 3593.61 (43.41) |
| | | FineTune | **5343.54 (4.65)** | 4423.00 (29.06) | 5336.47 (10.96) | 4573.23 (30.96) |
| | | Change | 265.57 | -242.41 | 1425.44 | 979.62 |
| | medium expert | Offline | 4750.21 (24.21) | 4763.72 (24.29) | 3787.70 (65.10) | 3403.58 (37.71) |
| | | FineTune | **5254.04 (19.83)** | 4599.00 (43.19) | 3983.42 (117.36) | 4596.49 (11.26) |
| | | Change | 503.83 | -164.72 | 195.72 | 1192.91 |
| | medium replay | Offline | 3391.06 (26.64) | 3427.12 (18.50) | 2410.17 (48.23) | 2161.59 (16.45) |
| | | FineTune | **4512.61 (4.01)** | 4269.35 (15.13) | 1752.88 (112.71) | 2673.63 (17.31) |
| | | Change | 1121.54 | 842.24 | -657.3 | 512.04 |
| | medium | Offline | 3637.89 (36.90) | 3642.84 (20.60) | 3211.47 (12.66) | 2606.69 (29.86) |
| | | FineTune | **4826.40 (1.55)** | 4266.09 (12.80) | -2103.99 (10.54) | 3132.12 (22.55) |
| | | Change | 1188.5 | 623.25 | -5315.46 | 525.43 |
| HalfCheetah | expert | Offline | 11347.06 (5.12) | 11089.65 (24.91) | 959.41 (12.87) | 11151.42 (4.46) |
| | | FineTune | 11196.82 (9.57) | **11408.80 (4.98)** | 6546.81 (33.36) | 11010.34 (4.78) |
| | | Change | -150.24 | 319.15 | 5587.4 | -141.08 |
| | medium expert | Offline | 10086.43 (42.11) | 10054.46 (56.34) | 4511.98 (36.84) | 6431.29 (47.99) |
| | | FineTune | **11262.88 (6.22)** | 11218.53 (5.95) | 10347.94 (20.20) | 6034.64 (40.49) |
| | | Change | 1176.45 | 1164.07 | 5835.96 | -396.65 |
| | medium replay | Offline | 5209.87 (2.66) | 5289.54 (3.22) | 5275.87 (1.27) | 4096.27 (24.64) |
| | | FineTune | **6635.32 (16.13)** | 6156.15 (12.87) | **6628.35 (9.20)** | 4021.84 (15.08) |
| | | Change | 1425.45 | 866.6 | 1352.48 | -74.42 |
| | medium | Offline | 5716.72 (2.29) | 5742.89 (2.02) | 4359.90 (52.57) | 3696.49 (44.49) |
| | | FineTune | **7639.17 (8.72)** | 6513.38 (16.75) | 7439.24 (11.91) | 5005.84 (2.67) |
| | | Change | 1922.45 | 770.49 | 3079.34 | 1309.35 |
| Hopper | expert | Offline | 3346.42 (12.48) | 2885.90 (24.07) | 1445.60 (31.53) | 2544.39 (29.50) |
| | | FineTune | **3517.17 (5.66)** | 2168.89 (33.35) | 2515.50 (37.96) | 2955.99 (16.71) |
| | | Change | 170.75 | -717.02 | 1069.9 | 411.61 |
| | medium expert | Offline | 3032.05 (22.29) | 1992.49 (31.03) | 733.59 (10.99) | 1635.83 (8.51) |
| | | FineTune | **3015.84 (33.73)** | 1568.84 (48.16) | 2878.82 (27.02) | 1769.68 (7.49) |
| | | Change | -16.21 | -423.65 | 2145.23 | 133.85 |
| | medium replay | Offline | 2975.21 (12.04) | 1942.17 (15.85) | 2294.38 (31.45) | 638.10 (6.33) |
| | | FineTune | 3160.03 (5.50) | 1965.99 (43.32) | **3318.32 (2.89)** | 806.58 (8.47) |
| | | Change | 184.82 | 23.82 | 1023.93 | 168.47 |
| | medium | Offline | 1945.62 (6.19) | 1897.93 (9.11) | 2340.96 (14.89) | 1609.48 (4.22) |
| | | FineTune | 2593.49 (11.20) | 2006.70 (20.02) | **3064.90 (8.81)** | 1637.37 (4.88) |
| | | Change | 647.86 | 108.76 | 723.94 | 27.89 |
| Walker2D | expert | Offline | 5076.38 (4.04) | 4725.18 (27.57) | 5061.92 (1.02) | 3644.09 (47.59) |
| | | FineTune | **5088.79 (1.54)** | 4852.04 (9.34) | 4922.00 (20.22) | 4984.13 (1.37) |
| | | Change | 12.41 | 126.86 | -139.92 | 1340.04 |
| | medium expert | Offline | 5006.62 (4.27) | 4410.62 (16.43) | 2361.18 (60.66) | 4120.01 (29.44) |
| | | FineTune | **5149.60 (1.53)** | 2036.23 (21.97) | 3858.85 (67.75) | 4314.14 (24.81) |
| | | Change | 142.98 | -2374.39 | 1497.67 | 194.13 |
| | medium replay | Offline | 3205.26 (26.23) | 3034.25 (24.45) | 3553.79 (14.45) | 789.40 (15.84) |
| | | FineTune | 4391.99 (4.74) | 3937.08 (23.09) | **4453.95 (4.48)** | 634.43 (10.68) |
| | | Change | 1186.73 | 902.83 | 900.16 | -154.97 |
| | medium | Offline | 3790.83 (7.16) | 3269.64 (24.45) | 600.84 (39.67) | 2474.75 (29.75) |
| | | FineTune | **4113.62 (3.28)** | 3650.39 (12.29) | 453.01 (45.64) | 3045.55 (12.63) |
| | | Change | 322.79 | 380.75 | -147.82 | 570.8 |

Figure 14: The performance changes during fine-tuning. This table reports the score before normalization. The number in bracket is the standard error. Performance was averaged over 10 random seeds.

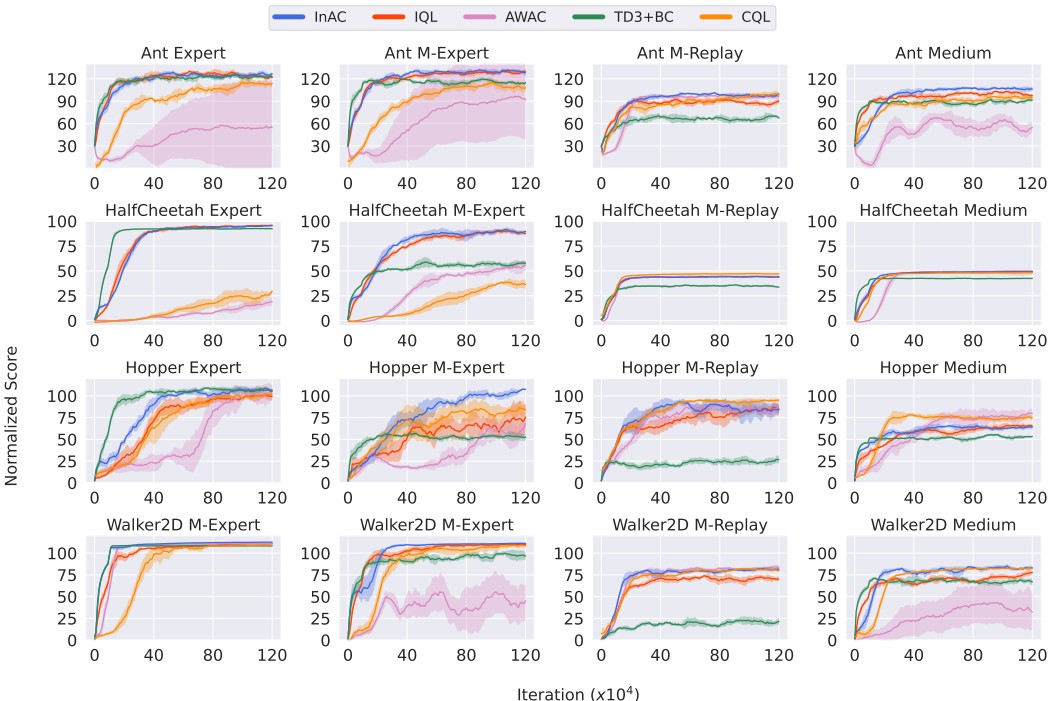

Figure 15: Offline learning curves with 1.2 million iterations. The x-axis is the number of iterations and the y-axis is the normalized score. Performance was averaged over 5 random seeds, after using a smoothing window of size 10.

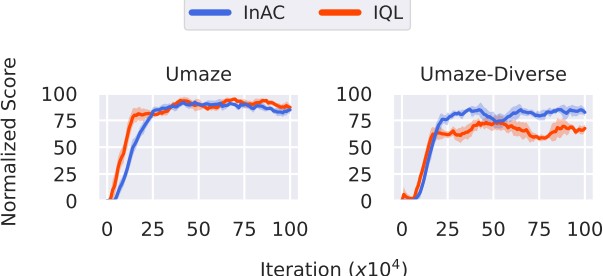

Figure 16: Offline learning curves with 1 million iterations. The x-axis is the number of iterations and the y-axis is the normalized score. Performance was averaged over 5 random seeds, after using a smoothing window of size 10.

