# OpenReview forum: "The In-Sample Softmax for Offline Reinforcement Learning"
_ICLR.cc/2023/Conference — ICLR 2023 notable top 25%_

### Official Review · Reviewer_xpgy · 2022-10-23

**Confidence:** 4
**Correctness:** 4
**Technical Novelty And Significance:** 3
**Empirical Novelty And Significance:** 2
**Recommendation:** 8

**Clarity, Quality, Novelty And Reproducibility:**

- Clarity: high. The paper is easy to read and to follow, takes time to explain everything needed to understand the algorithm, and does not have any distracting part.
- Quality: high. The proposed method is sound, easy to implement, cheap to compute, and compatible with state-of-the-art RL algorithms of the family of Soft Actor-Critic.
- Originality: high. The proposed algorithm is, to my knowledge, novel. Its derivation is interesting and novel too.

**Strength And Weaknesses:**

Strengths:

- The paper is very interesting to read and the idea is exciting! Some parts of the paper, that may appear simple (such as Equation 5, the trick to reduce the important of $\pi_D$), are eye-opening.
- The proposed algorithm is a relatively simple modification of the standard soft actor-critic losses. Nothing extra to compute, no resampling of the dataset to do. The algorithm seems easy to implement, and at almost no computational cost compared to vanilla SAC.
- The paper is very clear and well-presented. Just the right amount of maths is given to understand how the new losses are designed, why they make sense, and how they can be implemented. Equations 14 to 16 are nicely grouped together to give a direct overview of the full (modified) SAC algorithm.
- The sanity check is very nice to have, more papers should have a small and simple sanity check like this.

Weaknesses:

- The empirical evaluation is not as precise and complete as in other papers. Even in the appendix, I did not find absolute scores obtained by the agents in the environments (to be easy able to verify that the baseline scores match what other papers report, and to easily compare the scores obtained by IAC with baselines not present in the paper). Normalized scores are not that useful and usually hide more information than they reveal. Evaluation on AntMaze could also have been interesting: some people motivate that AntMaze is more difficult than the Mujoco tasks due to its sparse reward function, that better stresses the ability of an RL agent to bootstrap (instead of just following $r_t$).

**Summary Of The Paper:**

The paper proposes an Actor-Critic RL algorithm able to learn from an offline dataset, while addressing the problem of out-of-distribution actions. The core contribution of the paper is the identification of how nicely soft policies and Q-updates (based on the softmax over actions) are amenable to updates robust to out-of-distribution actions. The resulting algorithm looks like the Soft Actor-Critic, but with the actor and critic losses slightly modified to introduce a term that depends on the estimated behavior policy used to collect samples in the dataset. Experimental results show that the resulting algorithm outperforms several baselines on standard benchmarks, and a theoretical analysis explains why the algorithm makes sense.

**Summary Of The Review:**

Very well-written paper with sufficient (but can be improved) empirical results. I would strongly advocate for the authors to include raw (un-normalized) scores in the final version of the paper, for ease of comparison against algorithms not present in this particular paper.

---

> ### Author Response · Authors · 2022-11-15
> **Reply**
>
> We thank the reviewer for acknowledging our contribution and providing insightful suggestions. The detailed responses are provided below.
>
>
> **Absolute scores.**
>
> Thanks for pointing out this issue. We note that other works that use the D4RL dataset also use the normalized score to report the result. The difference is that previous works use the range 0-100 to report the normalized score, while we use the range 0-1. We agree that this makes it hard to compare with other baselines that are not present in the paper. To fix this, we have updated Figure 2, 3, 10, and 11 for offline learning results, and Figure 4 and 13 for fine-tuning results. We also add the absolute score for offline learning and fine-tuning (Figure 12 and 14 in Appendix) to facilitate comparisons to results reported in standard Mujoco benchmarks.
>
>
> **AntMaze.**
>
> Our key competitor, IQL, has not previously been run on sparse-reward Ant Maze. Instead, they ran it on a dense-reward version (see https://github.com/ikostrikov/implicit_q_learning/blob/09d700248117881a75cb21f0adb95c6c8a694cb2/train_offline.py#L68).
>
> We did run the comparison in the dense-reward Ant Maze setting they used. InAC had a better performance than IQL in d4rl’s antmaze-umaze-diverse-v0 dataset, and similar performance to IQL in antmaze-umaze-v0 dataset. We added the learning curve in Figure 16 in the Appendix.
>
> As for the sparse reward setting, we were not able to get IQL to work well, and so it is clearly an environment where we could see interesting differences! Thank you for this suggestion. As this will take some time to understand, and time to properly tune parameters, etc, as a next step, we will be investigating these methods in sparse-reward Ant Maze.

---

> > ### Public Comment · ~Yuwei_Fu1 · 2023-02-25
> > **IQL runs on sparse-reward Ant Maze.**
> >
> > IQL actually runs on the sparse-reward Ant Maze. The IQL agent only receives 0 score when reaches the goal otherwise it receives -1. The reward signal is shifted by 1 and is equivalent to the sparse-reward setting.

---

### Official Review · Reviewer_Dopc · 2022-10-25

**Confidence:** 3
**Clarity, Quality, Novelty And Reproducibility:** See above for more detailed comments.
**Correctness:** 4
**Technical Novelty And Significance:** 3
**Empirical Novelty And Significance:** 3
**Recommendation:** 6

**Strength And Weaknesses:**

Strength:
- The paper is well-written and easy to follow.
- Nice theoretical characterization.
- Experiments results seem promising.

Weaknesses/Questions:
- The idea is a bit straightforward (though not necessarily a weakness), as we have in-sample max to improve max, it seems not surprising to use in-sample softmax to replace softmax.
- The explanation/derivation in page 4 before section 4 may not be necessary as the math seems trivial. Also some argument could be more accurate (e.g. when $\pi_D(a|s) = 0$, equation (7) is undefined since $\log(\pi_D(a|s) )$ does not exist).
- The in-sample softmax optimal policy approaches the in-sample optimal policy as the temperature approaches zero. In practice, how do you choose the temperature. Have you considered annealing the temperature for better performance?

**Summary Of The Paper:**

In maximum entropy RL, we use a soft Bellman optimality equations where the standard max operation is replaced by the log-sum-exp operation. Just as researchers have proposed to use in-sample max instead of max over the entire action space, this paper propose to use in-sample softmax to improve the softmax operation, such that we only uses actions well-covered by the dataset. Some theoretical study are provided to show the convergence property similar to softmax. Experimental results demonstrate the effectiveness of the proposed in-sample softmax operation for a SAC-type algorithm on various offline RL benchmarks.

**Summary Of The Review:**

The paper propose to use in-sample softmax instead of softmax for maximum entropy offline RL, which leads to in-sample SAC that shows promising results on some offline RL benchmarks. The idea is interesting, though being a straightforward extension of existing ideas.

---

> ### Author Response · Authors · 2022-11-15
> **Reply**
>
> We thank the reviewer for the overall positive feedback and providing insightful suggestions. The detailed responses are provided below.
>
> **The idea is straightforward.**
>
> We agree that the in-sample softmax seems to be a simple idea at a first glance. However, we also believe that the significance of a contribution should be judged by whether it can solve major algorithmic/theoretical challenges in an elegant way. If the proposed method indeed matches this criteria, we believe “simple” should be a strength but not a weakness.
>
>
>
> **Equation 7 is undefined when $\pi_D(a|s)=0$.**
>
> Thanks for pointing out this problem. We would like to clarify that whenever $\pi_D(a|s)=0$, $\pi_{\pi_D, q} (a|s) = 0$. We have added a discussion in the revision to clarify the definition on the footnote of page 4.
>
>
> **Temperature**.
>
> For all experiments included, we swept the temperature in [1.0, 0.5, 0.33, 0.1, 0.01] and reported the best result (see Appendix B.4). In our results, temperature=0.1 and 0.01 were both good in most settings, with 0.01 notably better for the expert dataset. In our offline setting, we can pick a smaller temperature because we do not need to explore the space, so it is more reasonable to set it to one of these two smaller values. Of course, it could still have interesting interactions in terms of convergence (learning in fewer iterations on the offline dataset) and reaching a different solution. It is possible something like annealing could improve convergence, by starting at a higher temperature and then annealing to 0.01. We have not yet tested this idea, but we like the suggestion and will be trying it in future work.

---

### Official Review · Reviewer_LBrh · 2022-11-03

**Confidence:** 4
**Correctness:** 4
**Technical Novelty And Significance:** 3
**Empirical Novelty And Significance:** 3
**Recommendation:** 8

**Clarity, Quality, Novelty And Reproducibility:**

The paper is generally clear and well-written.

The in-sample softmax, while a combination of existing ideas, is novel to my knowledge. More importantly, the authors show identity (5), which is the core algorithmic innovation and addresses a key issue in offline RL.

The authors include detailed descriptions of the experimental setup and hyperparameters for reproducibility purposes.

A small note: in equation (47), I think the final equality is the wrong way. (The logic would be correct and match the statement of the lemma if the sign were flipped.)

**Strength And Weaknesses:**

Strengths:
* The paper presents a clean motivating theory that is not too far removed from the practical implementation.
* Unlike some other offline RL algorithms, InAC does not enforce constraints between the learned policy and the estimated behavior policy. This is good because imperfections in the estimated behavior policy can bias learning.
* Experiments show that the algorithm can effectively avoid OOD actions.
* The performance of the algorithm is similar to or better than existing algorithms, both for offline and offline-to-online learning.

Weaknesses:
* Judging by Figure 3, it seems that some of the algorithms have not converged within the 0.8M iterations. Obviously it is good that InAC learns faster, but it’s possible that the other algorithms could reduce the gap if given more iterations. Since training time is typically not the constraining factor in offline RL, I think all algorithms should be run to convergence.

**Summary Of The Paper:**

The paper proposes an offline RL algorithm based on the principles of in-sample maximization (when fitting the Q function) and maximum-entropy RL. It introduces the in-sample softmax, which is a straightforward modification of the usual softmax used in MaxEnt RL to only include actions in the support of the behavior policy, similarly to Batch-Constrained Q-Learning (BCQ). The convergence of in-sample soft policy iteration to an optimal policy (for an appropriate definition of optimal) is proved.

The main novelty is in showing that this in-sample softmax can be written as an expectation over samples from the dataset, reducing dependence on the behavior policy. Using this identity, the authors develop an algorithm similar to Soft Actor-Critic (SAC), which they dub In-sample Actor-Critic (InAC). Experiments demonstrate that InAC is effective in avoiding out-of-distribution (OOD) actions and attains high performance in both purely offline and offline-to-online settings.

**Summary Of The Review:**

I think the paper presents a useful idea and algorithm with compelling results, and therefore should be accepted.

---

> ### Author Response · Authors · 2022-11-15
> **Reply**
>
> We thank the reviewer for acknowledging our contribution and providing insightful suggestions. The detailed responses are provided below.
>
> **0.8 M iterations**
>
> We provide learning curves with 1.2 M training steps in Appendix B.1 (Figure 15). In our original experiment setup the mini-batch size was set to 100 (Appendix B.4). During the rebuttal period we find that using batch size 256 increases the performance of all algorithms in general. We thus use this new parameter and report the results after 1.2 M training iterations. The observations remain similar as our previous results: although the gap between IQL (the strongest baseline) and InAC reduces, our algorithm still performs best or nearly so across all domains. Due to the running time, we have not got the CQL results yet. We will add it to the camera ready version.
>
> **Equation 47**
>
> Thanks for pointing out this error. Yes, final equality is the wrong way. We have fixed this bug in the revision.

---

### Decision · Program_Chairs · 2023-01-20

**Decision:**

Accept: notable-top-25%

**Justification For Why Not Higher Score:**

The experiments are not as precise and complete as in previous papers.

**Justification For Why Not Lower Score:**

The proposed algorithm is simple and effective and have good empirical performance.

**Metareview: Summary, Strengths And Weaknesses:**

Summary:
This paper studies offline RL. The main novelty is showing that in-sample softmax can be written as an expectation over samples from the dataset, reducing dependence on the behavior policy. It proves the convergence of the in-sample soft policy iteration. Experimental results show that the resulting algorithm outperforms several baselines on standard benchmarks.

Strength:
-  Clean motivating theory,  theory inspired practical implementation.
-  no constraints enforced between the learned policy and the estimated behavior policy.
- Experiments show that the algorithm can effectively avoid OOD actions and have good performance.

Weakness:
- The empirical evaluation is not as precise and complete as in other papers.




**Note From Pc:**

if the above contains the word "oral" or "spotlight" please see: "oral" presentation means -> notable-top-5% and "spotlight" means -> notable-top-25%. As stated in our emails, we are disassociating presentation type from AC recommendations